# FROM CHEAP GEOMETRY TO EXPENSIVE PHYSICS: A PHYSICS-AGNOSTIC PRETRAINING FRAMEWORK FOR NEURAL OPERATORS

**Zhizhou Zhang**[1,*]**, Youjia Wu**[1]**, Kaixuan Zhang**[1]**, Yanjia Wang**[1,2]
[1]Bosch (China) Investment Co., Ltd.
[2]University of Hong Kong
{zhizhou.zhang, youjia.wu, kaixuan.zhang}@cn.bosch.com
ansleyw@connect.hku.hk

## ABSTRACT

Industrial design evaluation often relies on high-fidelity simulations of governing partial differential equations (PDEs). While accurate, these simulations are computationally expensive, making dense exploration of design spaces impractical. Operator learning has emerged as a promising surrogate for high-fidelity physical simulations, enabling fast prediction of partial differential equation (PDE) solutions. However, accuracy of neural operators is largely affected by the amount of training data, which must be generated by expensive numerical solvers. In practical industrial scenarios, there exists large collections of candidate geometries remain unsolved due to the high computation cost. These geometry-only samples contain no physical field labels and are therefore ignored in standard operator learning pipelines. In this work, we propose a general physics-agnostic pretraining framework to exploit this abundant geometric resource to improve the performance of neural operators. Specifically, we pretrain an autoencoder on a self-supervised proxy task to reconstruct geometry (e.g., via occupancy), learning an expressive latent representation without PDE supervision. Neural operators then leverage the pretrained latent embedding to learn more effectively from limited physics labels. An error decomposition analysis is provided to help understand the effectiveness of the physics-agnostic pretraining framework. Across four PDE datasets and three state-of-the-art transformer-based neural operators, our pretraining strategy consistently improves prediction accuracy. These results demonstrate that representations from physics-agnostic pretraining provide a powerful foundation for data-efficient operator learning. The code is publicly available at: https://github.com/zzzwoniu/Physics-agnostic-Operator-Pretraining.git

## 1 INTRODUCTION

Partial differential equations (PDEs) are fundamental descriptors for various physical and engineering systems (Guenther & Lee, 1996). Traditionally, PDEs are solved numerically by various discretization methods such as finite element, finite difference, finite volume, etc. (Reddy, 1993; Perrone & Kao, 1975) which offer high accuracy at high computational cost (Greenspan, 1955). This costly simulation process can be significantly accelerated by training surrogate neural operators to learn mappings from input functions to PDE solutions. As the pioneering work on neural operators, DeepONet adopted MLP architectures and established theoretical foundations for operator learning (Lu et al., 2021). Kernel-based operators (Li et al., 2020a;b) were then proposed to better handle resolution invariance. Recent studies focus on transformer-based neural operators (Cao, 2021; Hao et al., 2023; Wu et al., 2024) to handle complex irregular geometry inputs and further improve computational efficiency.

Despite architectural progress, the accuracy of neural operator still depends heavily on labeled PDE solutions produced by costly simulations. Considerable effort has therefore been devoted to improving data efficiency. For example, Cheng et al. (2024) predicts solution deviation relative to a

---

*Corresponding to zhizhou.zhang@cn.bosch.com

reference field to simplify the learning problem, and Deng et al. (2024) couples neural operators with a Point-BERT pretrained on ULIP-2 to inject geometric priors from large scale surface point clouds. However, these approaches assume particular problem structures, data modalities, or supervision sources, which can limit their general applicability.

On the other hand, operator pretraining has become an active research direction for enhancing PDE data efficiency. Most existing pretraining strategies are physics-aware and can be broadly grouped into two categories: The first pretrains models to reconstruct PDE solution fields to inject inductive bias (Serrano et al., 2024; Rahman et al., 2024). The second performs large-scale pretraining and fine-tuning over families of related PDE dynamics, often in autoregressive settings (Hao et al., 2024; Herde et al., 2024; McCabe et al., 2024). While effective, these pretraining techniques still rely on PDE solutions, which remain the primary computational bottleneck in industrial practice. By contrast, geometry-only data (meshes without solver labels) are plentiful yet underused. This physics-agnostic resource offers an untapped path to improve operator learning under label scarcity.

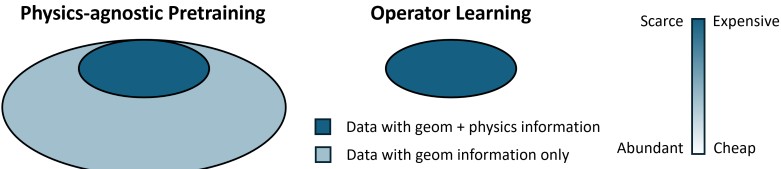

Figure 1: Data availability and usage in the proposed physics-agnostic pretraining framework. Pretraining leverages the abundant geometry-only data, while operator learning stage only observes limited PDE labeled samples.

We propose a physics-agnostic pretraining framework to exploit abundant geometric information while requiring PDE labels only during supervised operator training (Fig. 1). Specifically, we pretrain a point-cloud variational autoencoder (VAE) to extract latent geometry representations using a large point-cloud/mesh dataset. A physics-agnostic proxy task is introduced as the reconstruction objective for VAE training without requiring any PDE labels. After pretraining, neural operators are conditioned on these latent geometry embeddings rather than raw point clouds when predicting PDE solutions. As the pretraining stage is exposed to far more geometries than labeled PDE solutions, the encoder learns robust geometric structure and mitigates information loss from downsampled point clouds, yielding more informative representations and improving data efficiency on scarce PDE datasets. Our main contributions are:

- We introduce a novel physics-agnostic pretraining framework for operator learning tasks. Pretraining is performed on unlabeled point clouds via a proxy task to learn latent geometry representation. Neural operators are then trained to predict PDE solutions from these latents. This scheme allows full exploitation of geometry-only data without PDE labels.
- We introduce occupancy reconstruction as the objective of pretraining, allowing physics-agnostic self-supervised representation learning on irregular geometries realized as point clouds. The selection of proxy task is flexible, which can be swapped for signed distance fields, shortest vector fields, and other choices.
- We examine the proposed pretraining framework across four PDE problems and three transformer-based neural operators. Experiments show consistent performance gain when compared with direct single stage supervised training.

## 2 PRELIMINARIES

In this section, we define the problem and data settings. Then we briefly summarize related works on operator learning and pretraining.

### 2.1 PROBLEM SETUP

We consider PDE problems defined in the domain $\Omega \subset \mathbb{R}^d$ to be approximated with a learned operator $\mathcal{F} : A \to U$ where $A$ represents the input function space and $U$ represents the solution function space

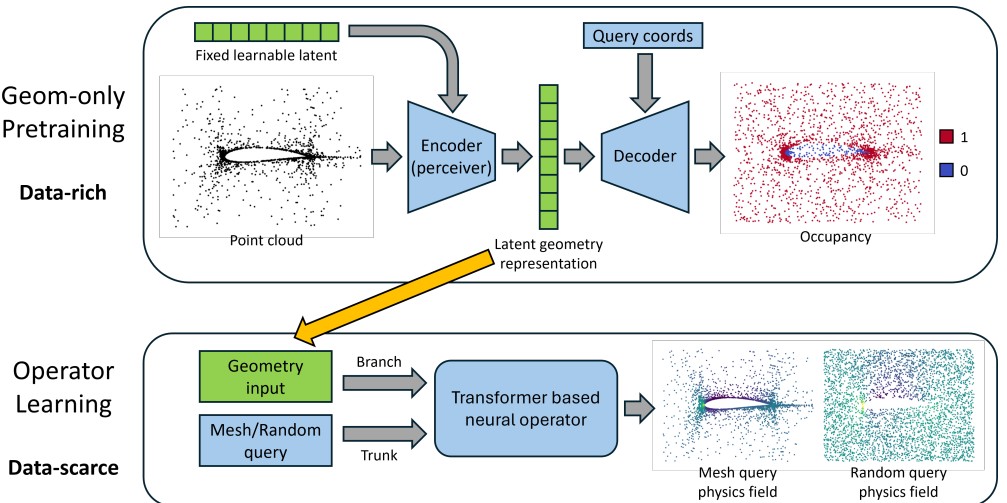

Figure 2: An illustration of our pretraining framework. We pretrain an encoder via a physics-agnostic proxy task (for instance the occupancy field) to extract information from rich geometry-only data. Unlike masked autoencoder, we use transformer-architecture to allow arbitrary query coordinates. Conditioning on the encoded embeddings, neural operators are trained with limited physics data to predict PDE solution fields.

of the physics over $\Omega$. In general scenarios, $A$ may consist of various types of information such as object geometries (material distribution), boundary conditions, source functions, etc. For simplicity, the input function space is assumed to contain only object geometry information, and the solution function is assumed to be scalar valued in this work. A realization of the geometry information $a_k \in A$ is represented as a point cloud (nodes of a mesh) $a_k = \{x_k^i\}_{i=1}^N = \mathbf{X}_k \in \mathbb{R}^{N \times d}$ with $N$ points in domain $\Omega$. And the physics solution function $u_k \in U$ is realized as $\{(y_k^i, u_k^i)\}_{i=1}^{N'}$ with $u_k^i = u_k(y_k^i)$, where $\{y_k^i \in \Omega\}_{i=1}^{N'}$ is a set of query points.

In this work, we consider two qualitatively different portions $D$ and $D'$ from each dataset:

$$D = \{(a_k, u_k)\}_{k=1}^{|D|}, \quad D' = \{(a_j)\}_{j=1}^{|D'|}, \quad |D'| \gg |D| \tag{1}$$

Here, $D$ is the physics portion with ground truth fields produced by a numerical solver, while $D'$ is the geometry-only portion that is abundant. This is highly related to many industrial scenarios where geometry designs are easy to generate, but physical field labels are limited. Motivated by such scenarios, we aim to leverage $D'$ to improve prediction accuracy on $D$, thereby alleviating the persistent data-scarcity bottleneck in operator learning.

## 2.2 RELATED WORK

**Neural Operators.** Operator learning aims to establish a mapping from input functions to solution functions. DeepONet established theoretical foundations for operator learning (Lu et al., 2021), followed by two important milestones: Graph Neural Operator (GNO) (Li et al., 2020b) and Fourier Neural Operator (FNO) (Li et al., 2020a). A set of follow-up works further extended GNO and FNO to irregular geometries (Li et al., 2023b;a) and improved computation efficiency (Guibas et al., 2021; Tran et al., 2021). With the rise of attention mechanism, architectural design shifted toward transformers (Cao, 2021; Li et al., 2022), where spatial locations and field values are tokenized as query, key, and value sequences. General Neural Operator Transformer (GNOT) Hao et al. (2023) introduced heterogeneous cross-attention to handle versatile input functions and gating mechanism to adjust network focus. Universal Physics Transformers (UPT) Alkin et al. (2024) adopts an encoder-decoder architecture to roll out physics evolution in the latent space, with an end-to-end training process. Transolver Wu et al. (2024) proposed the usage of physics attention to efficiently convert point based input tokens into learnable slices, which are projected back to query coordinates at the output layer. Latent Neural Operator (LNO) Wang & Wang (2024) proposed the invertible

physics-cross-attention to reduce the number of tokens in latent space. Overall, transformer-based neural operators have achieved state of the art in accuracy, offer greater input flexibility, and exhibit strong parallelization potential.

**Operator Pretraining.** Pretraining has been shown to be highly effective in natural language processing tasks Devlin et al. (2019); Raffel et al. (2020), computer vision tasks He et al. (2022); Dosovitskiy et al. (2020), and some scientific tasks Zhou et al. (2023); Nguyen et al. (2023). Existing operator pretraining work largely follows two paradigms. The first, exemplified by Rahman et al. (2024) and Serrano et al. (2024), pretrain latent representations via physical field reconstruction to enhance training stability and generalization. The second, represented by McCabe et al. (2024); Hao et al. (2024); Herde et al. (2024), performs large-scale autoregressive pretraining on spatiotemporal fields across datasets that share similar underlying fluid dynamics. While effective, both paradigms are inherently physics-aware and require expensive PDE solution labels, which is precisely the dominant computational bottleneck in many industrial workflows. In contrast, geometry-only data (meshes or point clouds without solver outputs) are often plentiful but underexploited. However, physics-agnostic pretraining remains relatively underexplored, with few existing attempts largely adapt masked autoencoder (MAE) strategies designed for regular grids and pixel space (Chen et al., 2024). This gap motivates the development of physics-agnostic pretraining methods that can directly leverage geometry-only datasets in unstructured settings.

## 3 METHODS

We propose to enhance the performance of neural operators via a physics-agnostic pretraining framework as seen in Fig. 2. The pretraining is performed on physics-agnostic $(a, o)$ data pairs from $D \cup D'$ to learn a better representation from input point clouds $a$. A proxy task $\mathcal{G} : A \rightarrow O$ can be obtained from $a$ at negligible computational cost, where $o$ is a field corresponding to geometric properties of point cloud $a$. The two datasets are therefore augmented to $D = \{(a_k, o_k, u_k)\}_{k=1}^{|D|}$ and $D' = \{(a_j, o_j)\}_{j=1}^{|D'|}$. A neural operator is then trained on $D$ with common supervised recipe to predict $u$ from the pretrained latent representation rather than from $a$.

### 3.1 PHYSICS-AGNOSTIC OPERATOR PRETRAINING VIA OCCUPANCY PREDICTION

To obtain a meaningful representation from pretraining without PDE solution labels, a proper proxy task should possess the following three features: **(1) Computation efficiency.** It should be broadly applicable and computable directly from the mesh/point cloud input $a$ at low computation cost. **(2) Consistency with operator learning.** It should stay under the umbrella of operator learning tasks to bridge naturally towards physical field prediction. **(3) Sampling invariance.** It should treat different point cloud realizations of the same underlying geometry equivalently as discussed in Appendix B. Ideally, the pretrained model would encode regions of high expected physical significance to better inform the neural operator to learn PDE solutions. However, in a fully general setting without explicit physics supervision, it is difficult to design an inductive bias that consistently improves performance across all problems. Taking all these factors into consideration, we pick occupancy field (Mescheder et al., 2019) $o \in O$ as the proxy task for the main experiments, which is realized (similar to $u$) by its values on a set of query points $\{(z_k^i, o_k^i)\}_{i=1}^{N''}$ with $o_k^i = o_k(z_k^i) \in \{0, 1\}$, where $o(z) = 1$ indicates that $z$ is inside an object and $o(z) = 0$ vice versa. Note that our framework can be easily extended to other proxy tasks. In section 4.4 we also explore signed distance fields (SDF) and shortest vectors (SV) (Jessica et al., 2023) as alternative proxy tasks, while reserving occupancy fields for the main experiments.

To perform pretraining on general geometry data via the occupancy reconstruction task, we follow the practice in Zhang et al. (2023) to train a point cloud variational autoencoder (VAE) as shown in Fig. 2. The encoder $\mathcal{E}$ adopts a perceiver architecture for aggregating the point cloud information into a fixed length vector, which is then projected into a probabilistic latent space via $MLP_\mu : \mathbb{R}^C \rightarrow \mathbb{R}^{C_0}$ and

$MLP_\sigma : \mathbb{R}^C \to \mathbb{R}^{C_0}$:

$$m^k = CrossAttn(\mathbf{L}, PosEmb(a_k = \mathbf{X}_k))$$
$$(h_\mu)_k^i = MLP_\mu(m_i^k)_{i \in [1,2,...M]} \tag{2}$$
$$(h_\sigma)_k^i = MLP_\sigma(m_i^k)_{i \in [1,2,...M]}$$

where $\mathbf{X} \in \mathbb{R}^{N \times d}$ is the $d$ dimensional point cloud of $N$ points, $PosEmb : \mathbb{R}^d \to \mathbb{R}^C$ is a positional embedding neural network, $\mathbf{L} \in \mathbb{R}^{M \times C}$ is a set of learnable tokens with fixed number $M$. The latent representation $z \in \mathbb{R}^{M \times C_0}$ is then sampled as:

$$h_k^i = (h_\mu)_k^i + (h_\sigma)_k^i \cdot \epsilon \tag{3}$$

where $\epsilon \sim \mathcal{N}(0,1)$. The decoder then predicts the occupancy value of a set of query points to reconstruct the geometry (occupancy field) $o_k(z_i) = \mathcal{D}(h_k)(z_i)$, so that the latent representation resides in function space (Appendix B). The training objective for the point cloud VAE takes the following form:

$$\min_{\phi,\eta} \frac{1}{|D \cup D'|} \sum_{k=1}^{|D \cup D'|} (\mathbb{E}_{z \sim p} \mathrm{BCE}(\tilde{\mathcal{D}}_\eta(\tilde{\mathcal{E}}_\phi(a_k))(z), o_k(z)) + \lambda \cdot \mathrm{KL}(\mathcal{N}(h_\mu, h_\sigma) \,\|\, \mathcal{N}(0,1))) \tag{4}$$

where $\mathrm{BCE}(\cdot, \cdot)$ stands for the binary cross entropy loss, and $\lambda$ is the KL weight. The sampling strategy $z \sim p$ for occupancy field consists of two parts: a uniform distribution over the computation domain $\mathcal{U}(\Omega)$, and perturbed point cloud $z^i = x^i + \varepsilon^i$ where $\varepsilon \sim \mathcal{N}(0, \zeta I)$ with $\zeta$ being a small positive scalar. Physics datasets are oftentimes sparse as they are computationally expensive to collect. However, the geometry dataset can be dense ($|D'| \gg |D|$) as the computation cost for generating new geometries is negligible.

## 3.2 Operator Learning

Instead of directly taking $a$ as inputs, neural operators $\mathcal{F}$ take advantage of the geometry representation learned from physics-agnostic pretraining (Fig. 2) in the form of $\tilde{\mathcal{F}}_\theta(\tilde{\mathcal{E}}_\phi(a_k)) = u_k$, where $\theta$ is a set of trainable parameters. Notice that the latent tokens produced by the pretrained encoder $\tilde{\mathcal{E}}_\phi$ can be smoothly integrated to transformer based neural operators discussed in section 2.2. The training goal is to minimize the relative L2 error, where $\hat{u}_k$ denotes the normalized ground truth physical field.

$$\min_\theta \frac{1}{|D|} \sum_{k=1}^{|D|} \frac{\sqrt{\sum_{i=1}^{N'} (\tilde{\mathcal{F}}_\theta^i(\tilde{\mathcal{E}}_\phi(a_k)) - \hat{u}_k^i)^2}}{\sqrt{\sum_{i=1}^{N'} (\hat{u}_k^i)^2}} \tag{5}$$

When training neural operators on physics data $D$, we only update parameters $\theta$ while keeping the encoder $\tilde{\mathcal{E}}_\phi$ frozen. In this work, we consider two types of different query strategies for $u$ that are widely used in various operator learning studies: directly querying the physical field on the point cloud (mesh) $\{y_k^i\} = \{x_k^i\}_{i=1}^N$, and querying the physical field according to a set of uniformly sampled random points $\{y_k^i \sim \mathcal{U}(\Omega)\}_{i=1}^{N'}$.

## 4 Experiments

We evaluate the proposed physics-agnostic pretraining framework on four different datasets as described in section 4.1 and compare with direct supervised learning results. The quantitative evaluation of our method is presented in sections 4.3 and 4.4, showing that training a latent geometry representation from rich but physics-agnostic data can enhance neural operator's understanding of physical fields.

## 4.1 Datasets

To examine our method, we modified the Elasticity (Li et al., 2023a) and AirfRans (Bonnet et al., 2022) datasets and built two new datasets to generate occupancy fields for physics-agnostic pretraining,

and PDE solution fields for operator learning. The geometries in $D'$ are sampled from the same (or similar) distribution as the geometries in $D$. We list the general statistics (Table 1) and the generation process of the datasets as the following. More details on datasets can be found in Appendix C.

**Stress.** We mimic the practice of the Elasticity dataset (Li et al., 2023a) to generate occupancy fields for physics-agnostic pretraining, and Von Mises stress fields for operator learning.

**AirfRans (near).** We directly adopt the pressure field data from AirfRans (Bonnet et al., 2022) and truncate the domain ($[-0.6, 1.6] \times [-0.5, 0.5]$) to focus only on the near volume of the airfoils. We then follow the instruction in (Bonnet et al., 2022) to generate new geometries via OpenFOAM and compute occupancy fields without running the CFD simulations.

**Inductor (3D).** We solve the Maxwell's equations using Comsol for a parameterized 3D iron core subject to the current excitation in a copper coil. The norm of the magnetic flux density field is exported as the physics field to be predicted. Occupancy fields are also generated for the parameterized iron cores.

**Electrostatics.** We solve the electrostatics Poisson equation in a domain that contains two materials with distinct permeability values. The two material domains are separated by a curved interface generated from a third order spline interpolation of 10 random points. We then compute the occupancy fields for physics-agnostic pretraining, and electric potential fields for operator learning.

Table 1: Statistics of the four datasets for the main experiments. The second row shows the size of physics data labeled with PDE solutions $u$, the third row shows the total data size including physics and geometry-only data, while the last two rows show the computation time for solving PDEs and mere mesh generation.

| Statistics | Stress | AirfRans (near) | Inductor (3D) | Electrostatics |
|---|---|---|---|---|
| Dimension | 2D | 2D | 3D | 2D |
| Size of $D$ (train+test) | 1900+200 | 800+200 | 900+300 | 1800+200 |
| Size of $D \cup D'$ (train+test) | 7600+500 | 2400+600 | 3600+300 | 7600+400 |
| Cost for $D$ (CPU hour) | 270 | 7680 | 1200 | 130 |
| Cost for $D'$ (CPU hour) | <0.1 | 0.14 | <0.1 | <0.1 |

All four datasets follow the format of the sample data point visualized in Figure 3. Physics-agnostic pretraining is performed on a combination of the two occupancy field sampling strategies on $D \cup D'$ as discussed in section 3.1. The neural operators are then trained on $D$ with the physics field prediction accuracy on random uniform queries and mesh-based queries reported separately. We also concatenate the occupancy value $o_k^i$ to the query coordinates $y_k^i$ as inputs for training neural operators, so that the PDE field data take the form $\{([y_k^i, o_k^i], u_k^i)\}_{i=1}^{N'}$. Note that for mesh-based queries where $\{y_k^i\} = \{x_k^i\}_{i=1}^{N}$, occupancy value is always 1.

## 4.2 MODELS AND TRAINING.

We benchmark three transformer-based neural operators GNOT (Hao et al., 2023), Transolver (Wu et al., 2024), LNO (Wang & Wang, 2024) by training each under the standard operator learning setting with and without our proposed physics-agnostic pretraining framework for performance comparison. The models are adjusted to have approximately the same number of learnable parameters. **Notice that we intentionally avoid exhaustive tuning of model or training hyperparameters, as our focus is on evaluating the effectiveness of the physics-agnostic pretraining strategy. Therefore, the reported performance should not be interpreted as a definitive comparison of model superiority.**

**Point cloud VAE.** We adopt the point cloud VAE architecture in Zhang et al. (2023) for physics-agnostic pretraining to extract latent embedding from geometry-only data, trained with loss function in Eq 4. We set the embedding dimension $C$ to 256, latent dimension $C_0$ to 32, learnable token number $M$ to 256, and depth of self-attention layers on latent space to 6.

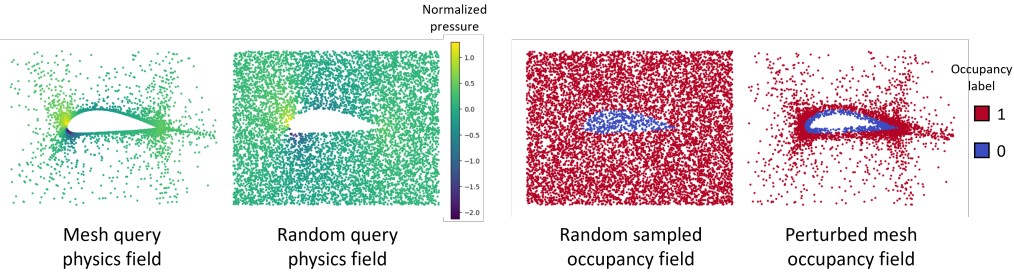

Figure 3: Visualization of a sample data point from the near volume AirfRans dataset. All of the datasets are preprocessed so that each data point consists of four types of information: physics field queried on the mesh point cloud, physics field queried on randomly sampled points, occupancy field queried on randomly sampled points, and occupancy field queried on the mesh point cloud perturbed by random displacements. **The geometry data $D'$ are labeled only with occupancy fields.**

**GNOT.** We adopt the GNOT model proposed in (Hao et al., 2023) as the first neural operator baseline, with a hidden size of 128, head number of 4, and layer number of 4 (each layer includes a linear self-attention and a linear cross-attention block). The gating strategy is excluded from the model to keep the comparison concise. The latent geometry representation can be fed directly into the GNOT branch net without modification.

**Transolver.** We adopt the Transolver model proposed in Wu et al. (2024) as the second neural operator baseline, with a hidden size of 128, head number of 4, slice number of 32, and 9 layers of physics-attention. The "get grid" layer is excluded from the model to keep the comparison concise. As the vanilla Transolver model assumes coupled query positions and point cloud/function observation positions, we replace the first physics-attention layer of Transolver with a linear cross-attention layer from GNOT to allow for latent geometry representation inputs.

**LNO.** We adopt the LNO model proposed in (Wang & Wang, 2024) as the third neural operator baseline, with a hidden size of 128, head number of 4, and 8 attention layers. The vanilla LNO assumes shared parameters for encoding position information of the trunk input and branch input. Instead, we use two distinct MLPs to allow for latent geometry representation inputs to the branch net.

**Training strategy.** In each pretraining iteration on VAE, 2048 points are sampled from the mesh to explicitly represent the geometry. 1024 points are sampled from the random occupancy field, and another 1024 points are sampled from the perturbed mesh occupancy field (illustrated in Figure 3) to estimate the geometry reconstruction error. We run 400 epochs using AdamW, with a half-cycle cosine scheduler of learning rate $1 \times 10^{-3}$ and minimum learning rate $1 \times 10^{-6}$. All pretraining processes of VAEs are completely agnostic of physics.

For all datasets, the first 100 physics data are used to normalize the physics field. We then compute relative L2 error (Eq 5) on the normalized physics field for training loss and evaluation metric. **Notice that the relative L2 error can appear much smaller on unnormalized datasets with large absolute values in the mean.** We run 200 epochs of training for experiments on neural operators, using the same optimizer and scheduler strategy as VAEs. In each operator training iteration, 2048 points are sampled from the mesh to explicitly represent the geometry, 4096 points are sampled from either the mesh query or random query physics field to estimate the operator prediction error. More details of GPU usage and compute resources are included in Appendix E.

### 4.3 MAIN RESULTS

We compare the performance of transformer-based neural operators between taking point clouds as inputs and taking latent embeddings from physics-agnostic pretraining as inputs. Table 2 shows the operator prediction error under mesh query and random query, with and without the pretrained

encoder. The VAEs and neural operators in the main experiments are trained with datasets listed in Table 1, with a fixed KL weight of 0.001.

Table 2: Comparison of prediction errors on the four datasets. We report the mean error, with the standard deviation from three independent trials shown in parentheses. The physics-agnostic pretraining framework is tested on three transformer-based neural operators and compared with results from standard operator learning. We use bolded font to highlight the results when pretraining enhances the performance of operator learning.

| Dataset | Query | Relative L2 on normalized data ($\times 10^{-2}$) | | | | | |
|---|---|---|---|---|---|---|---|
| | | GNOT | G+VAE | Trans | T+VAE | LNO | L+VAE |
| Number of parameters | | 1.7-1.8 M | | 1.7-1.8 M | | 1.8-1.9 M | |
| Stress | Mesh | 9.8(0.2) | **9.0(0.1)** | 11.5(0.2) | 11.2(0.2) | 26.5(1.3) | **13.6(0.6)** |
| | Random | 10.3(0.2) | **8.3(0.1)** | 11.5(0.6) | **9.7(0.2)** | 20.0(0.2) | **11.6(0.8)** |
| AirfR (near) | Mesh | 6.8(0.2) | **5.6(0.1)** | 13.4(0.3) | 12.7(0.3) | 27.4(1.3) | 27.1(0.2) |
| | Random | 7.8(0.2) | **5.9(0.1)** | 15.0(0.2) | **10.8(0.1)** | 25.3(1.8) | **10.0(0.2)** |
| Inductor (3D) | Mesh | 7.0(0.1) | 7.1(0.1) | 11.4(0.5) | **8.4(0.2)** | 24.9(1.4) | **9.2(0.6)** |
| | Random | 12.5(0.2) | **11.8(0.2)** | 16.8(0.6) | **13.2(0.2)** | 20.3(1.0) | **13.0(0.2)** |
| Electrostatics | Mesh | 4.2(0.1) | **3.3(0.1)** | 5.0(0.1) | **3.8(0.1)** | 13.5(0.4) | **4.6(0.2)** |
| | Random | 4.6(0.2) | **3.4(0.0)** | 5.6(0.1) | **3.9(0.1)** | 13.5(0.2) | **4.7(0.1)** |

Across most settings, we observe clear performance improvement on physical field prediction after introducing the physics-agnostic pretraining. The results are consistent across different datasets and neural operators, showing that the pretraining stage potentially constructs a stronger geometry representation than directly feeding raw point clouds. Note that point clouds are discretized approximations of real geometries, while the latent representation (residing in function space since it reconstructs the occupancy field) offers a more accurate approximation, especially after being enriched on diverse geometry data. We also observe larger improvements for randomly sampled query points than for mesh-based queries. One potential reason is that mesh-based sampling is denser than uniform sampling, particularly near geometry boundaries, so that baseline models readily capture critical geometric details, leaving less headroom for pretraining. This phenomenon is most pronounced on the 3D Inductor dataset, as uniform sampling in 3D space is especially inefficient.

## 4.4 ABLATION STUDY

**Number of geometry data.** As physics-agnostic pretraining is decoupled from PDE solution labels, it is important study the impact of geometry data amount $|D \cup D'|$ on operator performance. Experiments are conducted on the Stress dataset where we pretrain VAEs on geometries sampled from four types of different void priors as shown in Table 3. We then compare the performance of GNOT and Transolver using the latent space from each VAE as shown in Table 4. VAE1 is only exposed to the physics data $D$, while VAE2 and VAE3 are trained on different settings of geometry data $D'$. It can be observed that the latent representation becomes stronger as more geometry data are added, even though some are sampled from different distributions. The improvement is less significant when the latent representation isn't augmented by extra geometry data (GNOT/Transolver+VAE1). These observations verify our assumptions in Appendix B where the representation error can be reduced with increased number of unlabeled geometries.

Table 3: Metrics of pretrained encoders using different number of geometry data on the stress dataset. Details of shape priors are discussed in Appendix C.

| Stress data | VAE1 | VAE2 | VAE3 |
|---|---|---|---|
| Ellipse $a = 0.3$, $b = 0.15$ | 1900 | 3800 | 1900 |
| Ellipse $a = 0.15$, $b = 0.3$ | 0 | 0 | 1900 |
| Rectangle $a = 0.3$, $b = 0.15$ | 0 | 0 | 1900 |
| Rectangle $a = 0.15$, $b = 0.3$ | 0 | 0 | 1900 |
| IOU(%) | 99.6 | 99.8 | 99.8 |
| KL loss | 0.70 | 0.69 | 0.955 |

Table 4: Performance of neural operators using encoders pretrained on different number of geometry-only data.

| Dataset | Query | Relative L2 on normalized data ($\times 10^{-2}$) | | | |
|---|---|---|---|---|---|
| | | GNOT | G+VAE1 | G+VAE2 | G+VAE3 |
| Stress | Mesh | 9.8 | 9.9 | 9.7 | **9.0** |
| Ellipse $a = 0.3$, $b = 0.15$ | Random | 10.3 | 9.1 | 8.6 | **8.3** |
| 1900 | | Trans | T+VAE1 | T+VAE2 | T+VAE3 |
| | Mesh | 11.5 | 12.0 | 11.4 | **11.2** |
| | Random | 11.5 | 10.4 | 9.8 | **9.7** |

**Effect of KL weight.** We investigate the effect of KL weight ($\lambda$ in Eq 4). We train three VAEs and one deterministic autoencoder using all geometries in the Stress and the AirfRans datasets (Table 1). We observe higher accuracy when relaxing the KL weight, but all autoencoders achieve more than 99.5% IOU. The prediction accuracy of GNOT and Transolver using the encoded latent space of the four autoencoders are listed in Table 6. In most scenarios, the performance of neural operators are improved with the latent representation provided by the pretrained encoder. Although all four autoencoders achieve high reconstruction accuracy, using a probabilistic latent space in pretraining with a smaller regularization weight $\lambda = 0.001$ or $\lambda = 0.0001$ tends to provide more robust performance. That said, the optimal choice can depend on the specific neural operator architecture, as well as training hyperparameters, and the dataset being learned.

Table 5: Training metrics of point cloud autoencoder on the Stress and AirfRans (near) dataset at different KL weights.

| | KL weight | 0.01 | 0.001 | 0.0001 | Deterministic |
|---|---|---|---|---|---|
| Stress | IOU(%) | 99.5 | 99.7 | 99.8 | 99.8 |
| | KL loss | 0.19 | 0.98 | 2.94 | N/A |
| Airf (near) | IOU(%) | 99.9 | 99.9 | 99.9 | 99.9 |
| | KL loss | 0.06 | 0.36 | 1.62 | N/A |

Table 6: Prediction error of neural operators using latent geometry representation from VAEs pretrained with different KL weights. Here, AE stands for deterministic autoencoder.

| Dataset | Query | Relative L2 on normalized data ($\times 10^{-2}$) | | | | |
|---|---|---|---|---|---|---|
| | | GNOT | +VAE(0.01) | +VAE(0.001) | +VAE(0.0001) | +AE |
| Stress | Mesh | 9.8 | 9.9 | **9.0** | 9.3 | 10.3 |
| | Random | 10.3 | 9.7 | 8.3 | **8.0** | 8.9 |
| Airf (near) | Mesh | 6.8 | 6.6 | 5.6 | **5.3** | 5.9 |
| | Random | 7.8 | 6.2 | 5.9 | **5.2** | 5.9 |
| | | Trans | +VAE(0.01) | +VAE(0.001) | +VAE(0.0001) | +AE |
| Stress | Mesh | 11.5 | 12.1 | **11.2** | 11.3 | 11.7 |
| | Random | 11.5 | 11.4 | 9.7 | **9.4** | 10.4 |
| Airf (near) | Mesh | 13.4 | 13.2 | 12.7 | 12.3 | **11.3** |
| | Random | 15.0 | 11.1 | 10.8 | 11.1 | **10.3** |

**Alternative proxy tasks.** While occupancy fields are selected as the default proxy task in the main experiments, our proposed framework is plug-and-play with alternative proxy task choices. As discussed in section 3, the proxy task can be chosen flexibly as long as it doesn't require PDE solving (computationally cheap), and can be expressed as a field. For general adaptability, no prior knowledge of the governing PDE should be assumed. Under these constraints, suitable candidates are typically derived from material distribution fields or geometric representation fields. In CFD problems such as AirfRans, SDF and SV are widely considered informative because they explicitly encode the relationship between arbitrary spatial locations and the nearest boundary, which is the dominant geometric factor in many CFD problems (Jessica et al., 2023). Therefore, we adopt SDF or SV as the

proxy task in the physics-agnostic pretraining stage and evaluate the pressure prediction accuracy of GNOT on AirfRans as shown in Table 7. When compared with direct supervised operator learning, our proposed physics-agnostic pretraining approach remains effective in most scenarios, except for the SDF case on mesh-based physics field query, where the pretraining stage offers no measurable gain. Notably, using SV as the proxy task yields the best physics field prediction accuracy. These results demonstrate both the effectiveness and breadth of our framework beyond occupancy fields. However, SDF or SV are not necessarily as informative for PDE problems beyond CFD, especially when volumetric properties from the body mesh dominate over surface proximity. Therefore, we still believe that occupancy is a more general, intuitive, and task-agnostic proxy choice.

Table 7: GNOT performance on the AirfRans dataset: our physics-agnostic pretraining framework with occupancy, SDF, and SV proxy tasks vs. the standard operator learning results (also with SDF or SV included in inputs).

| Query | Relative L2 on normalized data ($\times 10^{-2}$) | | | | | |
|---|---|---|---|---|---|---|
| | GNOT | G+SDF | G+SV | G+OCC_VAE | G+SDF_VAE | G+SV_VAE |
| AirfRans (Mesh) | 6.8 | 5.2 | 5.4 | 5.6 | 5.3 | **4.8** |
| AirfRans (Random) | 7.8 | 6.5 | 5.1 | 5.9 | 5.4 | **4.9** |

## 5  CONCLUSION AND LIMITATION

The paper explores the feasibility of leveraging the abundant geometry data resource to augment the performance of neural operators on scarce physics labels. Specifically, we introduce a physics-agnostic pretraining framework that leverage geometry data via an occupancy field reconstruction proxy task. The pretraining stage can benefit from rich geometry data as it doesn't require PDE solutions which are the major computation bottleneck in operator learning tasks. The latent representation produced from the pretrained encoder is then integrated to transformer-based neural operators for supervised physical field prediction learning. Incorporation of physics-agnostic pretraining yields consistent accuracy gains across different datasets and neural operator backbones. The framework is also adaptable to different proxy tasks and operator architectures. The results highlight the importance of properly selecting and preprocessing the representation of geometry inputs to neural operators.

The proposed physics-agnostic pretraining also possesses a few limitations. First, the effectiveness of occupancy reconstruction with multiple interacting geometries remains unexplored. Such settings may require replacing the BCE objective with multi-class cross entropy loss. Second, the default choice of occupancy is intuitive. Although small ablation studies on SDF and SV proxy tasks are included, a more systematic study is needed to help identify the most effective physics-agnostic proxy task based on the governing physics, input geometry format, and data accessibility. Lastly, integration between stages could be deepened beyond simply feeding pretrained latent embeddings. Potential strategies include joint or sequential finetuning with adjusted learning rates, hybrid losses that couple reconstruction and PDE supervision, or adapter-based conditioning.

## REPRODUCIBILITY STATEMENT

Detailed descriptions of the experimental setup, task definitions, and evaluation metrics are provided in section 4 and Appendix C. The code is publicly available at: https://github.com/zzzwoniu/Physics-agnostic-Operator-Pretraining.git

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

## A   LLM USAGE

LLM is only used on grammar polishing for this paper.

## B   PHYSICS-AGNOSTIC PRETRAINING: ERROR DECOMPOSITION AND LATENT CHART VIEW OF SHAPE SPACE

**Error decomposition.**   To help understand how physics-agnostic pretraining can improve downstream operator learning, we analyze the prediction error from neural operators. Let $N_{\text{geo}}$ denote the number of total geometries (including both labeled and unlabeled) used in pretraining stage and $N_{\text{phy}}$ the number of physics-labeled samples used in supervised operator learning stage. The composed predictor takes the form $F_\theta \circ \mathcal{E}_\phi$, where $\mathcal{E}_\phi$ is the pretrained geometry encoder and $F_\theta$ is the neural operator trained on physics data. Under standard regularity assumptions for representation learning and empirical risk minimization, the prediction error of the composed predictor can be decomposed into three major sources as the following:

$$\text{Error}(F_\theta \circ \mathcal{E}_\phi) \; \lesssim \; \underbrace{\mathcal{O}\left( \frac{C_{\mathcal{E}_\phi}}{\sqrt{N_{geo}}} \right)}_{\textbf{Representation error}} + \underbrace{\mathcal{R}_{N_{phy}}(F_\theta) + \mathcal{O}\left( \frac{1}{\sqrt{N_{phy}}} \right)}_{\textbf{Estimation error}} + \underbrace{\varepsilon_{approx}}_{\textbf{Approximation error}} . \quad (6)$$

Where the representation error term quantifies how well physics-agnostic pretraining produces a geometry representation. This term can be critical as point clouds are only digital approximations (discrete sampling) to the actual geometry. Ideally, the representation would be relevant to the downstream operator learning task, which is difficult under the physics-agnostic setting where no prior knowledge is available. When the pretraining stage is well designed, the constant $C_{\mathcal{E}_\phi}$ becomes smaller, indicating smoother or more structured input domain for the PDE operator. The estimation error term is the standard error from learning $F_\theta$ with finite $N_{phy}$ physics data. The third term $\varepsilon_{approx}$ is the inherent approximation limitation of the selected operator architecture.

**Reduce representation error by physics-agnostic pretraining.**   In canonical settings, neural operators can only learn from data that are labeled with PDE solutions: $N_{geo} = N_{phy}$. However, the effort of expanding $N_{geo}$ is significantly lower than obtaining $N_{phy}$ in industrial scenarios due to expensive simulations. Eq 6 points out a potential performance improvement of operator learning given fixed number of physics data: we can tighten the overall bound by reducing the representation error term via the following two routes: (i) designing a good proxy task so that $C_{\mathcal{E}_\phi}$ is small (latent features from strong geometry representation and align well with field prediction tasks), and (ii) leveraging abundant geometry-only data ($N_{geo} \gg N_{phy}$) to improve the bound at rate $\mathcal{O}(1/\sqrt{N_{geo}})$. Empirically, these predictions match our observations: incorporating physics-agnostic pretraining and scaling the amount of geometry data yields consistent improvements on downstream field prediction accuracy.

**Design principles for physics-agnostic pretraining.**   As physics-agnostic data mostly consist of pure geometry information, we choose to follow the central objective in shape analysis and geometry processing, which is to represent geometric objects while discarding nuisance variations such as orientation, sampling density, or point ordering. This naturally motivates organizing point clouds into equivalence classes: two point clouds are considered equivalent if they represent the same underlying shape up to transformations such as rigid motion, reparameterization, or sampling artifacts. The set of equivalence classes forms a moduli space: a space whose points correspond to distinct geometric structures modulo the chosen equivalence relation. In this sense, learning a geometry representation corresponds to coordinatizing the moduli space of shapes.

An autoencoder provides a data-driven approach to this coordinatization where the encoder acts as a learned coordinate chart, mapping each point cloud to a latent vector that represents its equivalence class, while the decoder ensures that this latent code retains enough information to reconstruct the original geometry. Thus, the latent space can be interpreted as a learned, low-dimensional chart on the moduli space of point cloud geometries. This can be illustrated by the t-SNE plot in Figure 4. Point clouds corresponding to the same geometry form tight, well-separated clusters, indicating

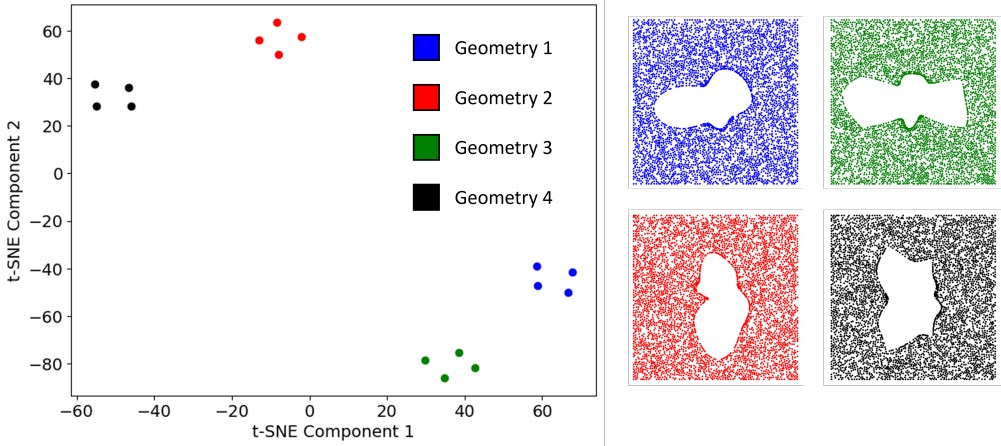

Figure 4: Visualization of pretrained VAE latent space via t-SNE. The plot shows latent representations of 4 different geometries; for each geometry, 4 different point clouds are sampled and encoded. Point clouds from the same geometry stay closer in latent space, consistent with a learned chart on shape space.

that the encoder learns a meaningful representation in which distances in latent space correlate with differences between underlying geometries.

**Formal description.** Let $\mathcal{X} \subset \mathbb{R}^{N \times d}$ denote the space of point clouds with $N$ points in $\mathbb{R}^d$. Define an equivalence relation $\sim$ such that $P_1 \sim P_2$ if they represent the same underlying shape up to transformations from a group $G$ (e.g., permutation, sampling/density variation, and optionally rigid motion via augmentation). The associated moduli space is the quotient space

$$\mathcal{M} = \mathcal{X}/\sim,$$

where each point $[P] \in \mathcal{M}$ represents an equivalence class of point clouds corresponding to a single geometric structure.

An autoencoder consists of an encoder $\mathcal{E} : \mathcal{X} \to \mathbb{R}^k$ and a decoder $\mathcal{D} : \mathbb{R}^k \to \mathcal{X}$ trained to minimize a reconstruction loss

$$\mathcal{L}(P) = \text{Dist}(P, \mathcal{D}(\mathcal{E}(P))),$$

where Dist is a distance metric invariant under the group $G$. Ideally, the encoder should be invariant under $\sim$, i.e., $P_1 \sim P_2 \Rightarrow \mathcal{E}(P_1) = \mathcal{E}(P_2)$, implying that it factors through the projection $\pi : \mathcal{X} \to \mathcal{M}$. This yields the commutative diagram:

$$
\begin{array}{ccc}
\mathcal{X} & \xrightarrow{\quad \mathcal{E} \quad} & \mathbb{R}^k \\
& \searrow_{\pi} \quad \nearrow_{\varphi} & \\
& \mathcal{M} &
\end{array}
$$

where $\mathcal{E} = \varphi \circ \pi$. Although $\pi$ is not explicitly computed in practice, this formalization clarifies the role of physics-agnostic pretraining: the encoder learns a data-driven coordinate chart on the moduli space of shapes, producing a structured latent domain that supports a smaller representation constant $C_{\mathcal{E}_\phi}$ for downstream operator learning.

## C  DATASETS

In this section, we explain the details of the datasets including the governing equations, boundary conditions, material property, and geometry configurations. The original AirfRans dataset (Bonnet et al., 2022) includes detailed airfoil boundary shape information for computing occupancy fields and generating new geometries. However, we choose to reproduce the Elasticity dataset (Li et al., 2023a) as the detailed void geometry information is missing.

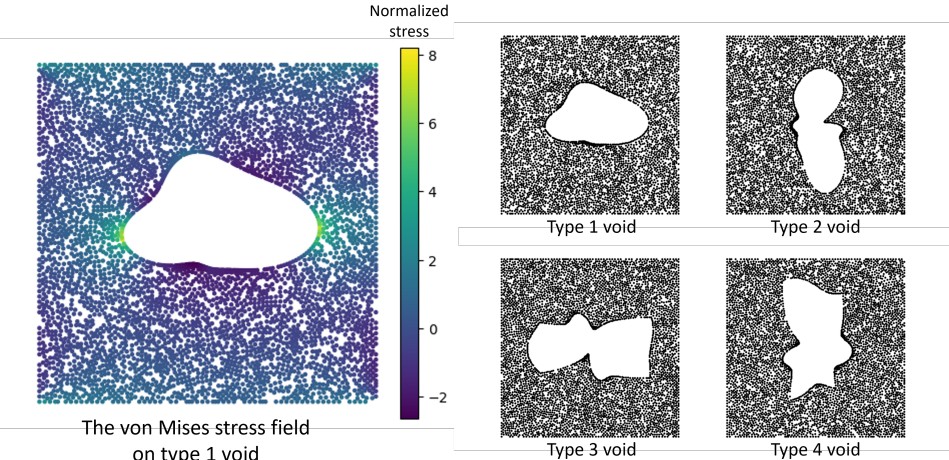

Figure 5: Examples of physics data and geometry data from the Stress dataset. Neural Operators only learn from von Mises stress fields on type 1 void, while observing the geometry data from geometries generated from 4 types of different void priors.

**Stress.** We mimic the Elasticity dataset in (Li et al., 2023a) and regenerate the geometries and stress fields in Fenics. We find the distribution of von Mises stress $\sigma_{vm}$ in a domain with an irregular void inside with a constant displacement vector at the top. We solve the following PDE:

$$\nabla \cdot \boldsymbol{\sigma}(u) = 0 \tag{7}$$

where $u(x)$ is the displacement vector field, $\boldsymbol{\epsilon} = \frac{\partial u}{\partial x}$ is the strain tensor, and $\boldsymbol{\sigma} = \frac{\partial w(\boldsymbol{\epsilon})}{\partial \boldsymbol{\epsilon}}$ is the stress tensor. To obtain the energy $w(\boldsymbol{\epsilon})$ we need to start with Cauchy Green stretch tensor $\boldsymbol{C} = 2\boldsymbol{\epsilon} + \mathbb{I}$, and then get $w(\boldsymbol{\epsilon}) = C_1(\text{tr}(\boldsymbol{C}) - 3) + C_2(\frac{1}{2}(\text{tr}(\boldsymbol{C})^2 - \text{tr}(\boldsymbol{C})))$. Two energy density parameters are: $C_1 = 1.863 \times 10^5$ and $C_2 = 9.79 \times 10^3$. One can then plug this result back into Equation 7 to get an equation in terms of $u(x)$. Note that although Equation 7 is linear in terms of $\boldsymbol{\sigma}$, it is highly non-linear in terms of actual underlying field variable $u(x)$. After obtaining the displacement vector field, one can calculate stress tensor $\boldsymbol{\sigma}$, and then get von Mises stress field $\sigma_{vm}$ from components of $\boldsymbol{\sigma}$:

$$\sigma_{vm} = \sqrt{\sigma_{xx}^2 - \sigma_{xx}\sigma_{yy} + \sigma_{yy}^2 + 3\sigma_{xy}^2} \tag{8}$$

For the solution domain, we have a square domain of $[0, 1] \times [0, 1]$ with an irregular hole at the center. This hole is described by an expression in polar coordinates: $r = r(\theta)$. In our setup, $r(\theta) = f(a, b, \theta) + \text{GRF}(A, l, \theta)$. The first $f$ term represents the prior void shape. The second term represents a Gaussian random field with an amplitude of $A = 0.05$ and correlation length of $l = 0.6$. We generate geometries using 4 types of different void priors as listed in Table 3, where $a$ represents the horizontal axis/edge of ellipse/rectangle and $b$ represents the vertical axis/edge of ellipse/rectangle. Example of the solution $\sigma_{vm}$ and geometries from different void priors are shown in Figure 5. We generate 1900 training data and 100 testing data (occupancy fields) for each type of void for geometry representation learning. Neural operators are trained with 1900 data and tested with 200 data (von Mises stress fields) from type 1 void only.

**AirfRans (near).** We directly adopt the simulation data in AirfRans Bonnet et al. (2022) which solves the Reynolds-Averaged Navier-Stokes Equations for airfoils. Unlike the other three datasets, AirfRans (near) contains two global features: attack velocity and angle, which are directly concatenated to all queries. The original dataset consists of 800 training data and 200 testing data with both geometry and physics information. We train transformer-based neural operators to predict the pressure fields around the airfoils in a truncated domain $[-0.6, 1.6] \times [-0.5, 0.5]$ as shown in Figure 3. To learn the geometry representation, we follow the same airfoil geometry parameterization (with a different random seed) in AirfRans and generate 2000 geometry only datasets (occupancy fields) with 1600 for training the VAE, and 400 for testing.

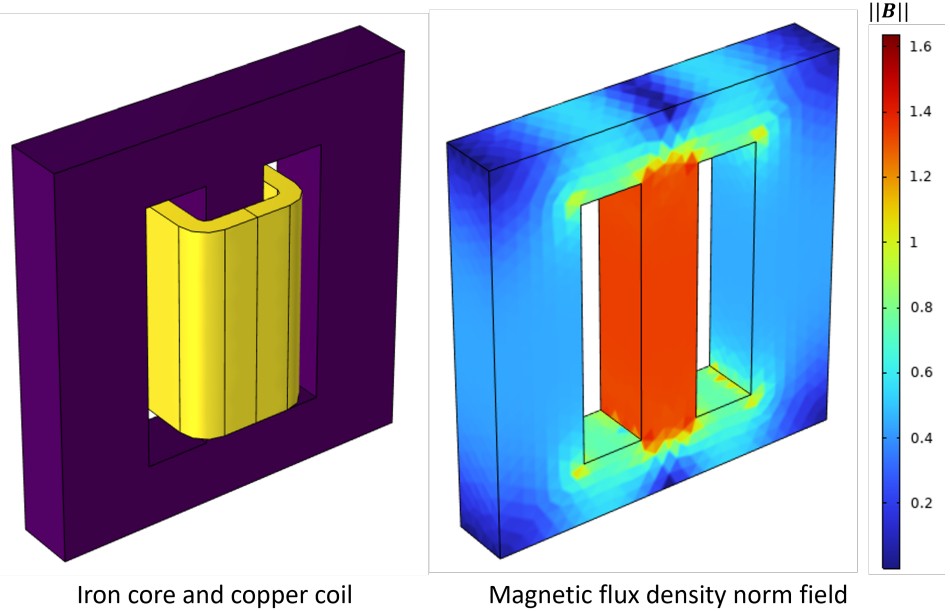

Iron core and copper coil          Magnetic flux density norm field

Figure 6: Example of a 3D inductor data. We generate the geometries (iron core in purple, coil in yellow) by parameterizing an EE type iron core, and calculate the corresponding magnetic flux density norm field (on the right).

**Inductor 3D.** We calculate the magnetic flux density field $B$ around a 3D inductor by solving the Maxwell's equations (Eq 9) in the frequency domain using Comsol.

$$\nabla \cdot B = 0$$
$$\nabla \times H = J + j\omega\epsilon E \qquad (9)$$
$$\nabla \times E = -j\omega\mu H$$

where $H$ stands for magnetic field intensity, $E$ is the electric field, $J$ is the current density field, $\epsilon$ is material permittivity (which is assumed to be 1 everywhere), and $\mu$ is material permeability. Figure 6 shows an example of the 3D inductor, consisting of an EE type iron core whose mid pillar is surrounded by a copper coil within a computation domain of $[-70, 70] \times [-70, 70] \times [-70, 70]$ $mm^3$ (normalized for VAE and neural operator training). Magnetic insulation $n \times A$ is applied to the boundary of the computation domain, where $n$ stands for the unit normal and $A$ is the magnetic vector potential satisfying $B = \nabla \times A$. The copper coil contains 500 turns with a current density of $1e6$ $A/m^2$.

The iron core follows a nonlinear magnetizing curve as shown in Figure 7. The neural operators are trained to predict the steady state magnitude of the magnetic flux density field $||B||$ at a high frequency of $f = \frac{\omega}{2\pi} = 300$ kHz. The nonlinear material property curve and high excitation frequency make this 3D inductor dataset extremely challenging. We generate 3600 training data and 300 testing data by randomly selecting the geometry parameters of the iron core. We compute occupancy fields for all the data to train the VAE, while only 900 of the training data are simulated to train the neural operators as listed in Table 1.

**Electrostatics** An electrostatics dataset is built using Fenics where we solve for the electric potential $\phi(x)$ from the following equation:

$$-\nabla \cdot (\epsilon(x)\nabla\phi(x)) = \rho(x) \qquad (10)$$

where $\epsilon(x)$ is the permittivity field, and $\rho(x)$ is the charge density field. Both of these fields are scalar fields. Here we follow the Heaviside-Lorentz units, so permittivity is unitless. In our dataset, the whole solution domain $\Omega$ is $[0, 1] \times [0, 1]$ is divided into two parts separated by a random interface. The random interface between these two parts is determined with 10 random points. These 10 points

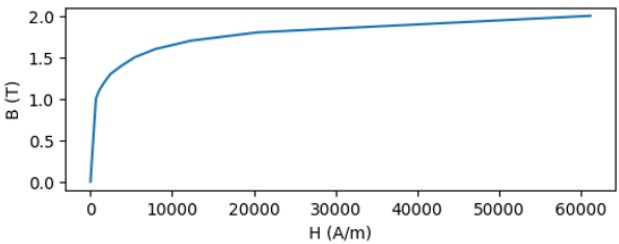

Figure 7: B-H curve of the iron core material.

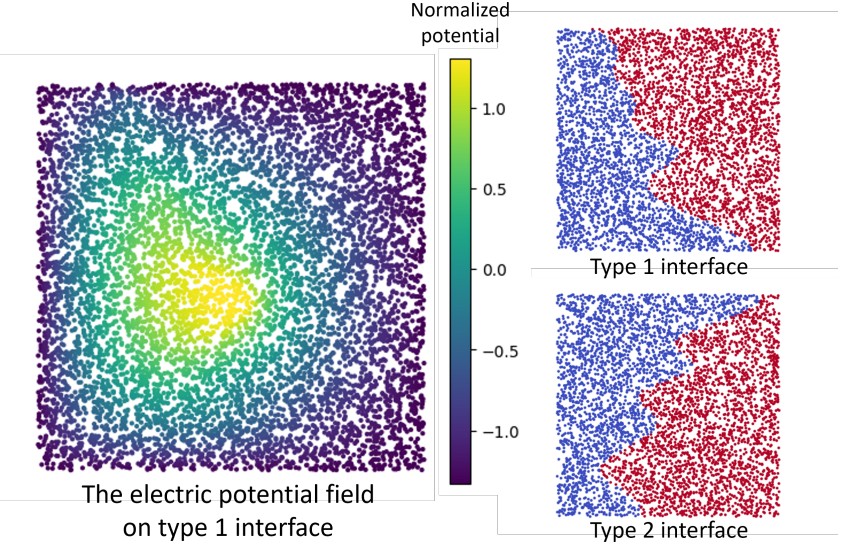

Figure 8: Examples of physics data and geometry data from the Electrostatics dataset. Neural Operators only learn from electric potential fields on type 1 interface, while observing geometry data generated from 2 types of different material interface priors.

are interpolated with third order spline interpolation to produce a curve as shown in the right part of Figure 8. The locations of the 10 random points are generated by adding a horizontal perturbation $\mathcal{U}(-0.2, 0.2)$ to 2 types of prior locations: 10 equally distanced points from $(0.3, 1)$ to $(0.7, 0)$ and from $(0.7, 1)$ to $(0.3, 0)$. A charge density of 1 and permittivity of 15 is present in the blue part, while a charge density of 0 and permittivity of 1 is in the red part. The solution of this example case is shown in the left part of Figure 8. We generate 3800 training data and 200 testing data (occupancy fields) for each type of interface for geometry representation learning. Neural operators are trained with 1800 data and tested with 200 data (potential fields) from type 1 interface only.

## D  ADDITIONAL EXPERIMENTS

In this section, we present a few additional ablation studies to examine the effectiveness of the proposed latent geometry representation under different data, training, model selection, and hyperparameter settings.

**Performance of UPT.**  We also evaluate UPT on the Stress and Electrostatics dataset using our framework with the results shown in Table 8. We implement a simplified version using transformer for encoder, decoder, and propagator, while omitting the message passing layer and inverse encoding decoding training to avoid extensive tuning. The results indicate that our framework can be smoothly adapted to UPT and improve its prediction accuracy on scarce physics labels. We notice that GNOT and Transolver (query at input layer) show superior performance compared to LNO and UPT (query

at output layer). A likely reason is that UPT and LNO are tailored for latent temporal rollouts, whereas all four benchmarks here are stationary physics. We also expect further improvement in UPT with the complete architecture and training recipe.

Table 8: Prediction error of UPT (simplified implementation) on the Stress and Electrostatics datasets. The physics-agnostic pretraining framework enhances the performance of UPT.

| Model | Relative L2 on normalized data ($\times 10^{-2}$) | | | |
| --- | --- | --- | --- | --- |
| | Stress, Mesh | Stress, Random | Elec, Mesh | Elec, Random |
| UPT | 26.3 | 20.8 | 11.6 | 12.0 |
| UPT+VAE | 12.8 | 10.6 | 4.9 | 4.8 |

**Scarce physics dataset.** To assess the effectiveness of physics-agnostic pretraining, we conduct experiments on the AirfRans dataset in a data-scarce setting. In this setting, only 200 physics samples with PDE solutions $u$ are used for training neural operators. The VAE is pretrained on the complete geometry dataset with 2400 airfoil point clouds. The results in Table 14 show that prediction accuracy decreases compared to full-data setting in the main experiments, as expected. Meanwhile, we observe clear performance gains after adopting the pretrained encoder in most cases (except for LNO under random query). This supports our intuition that physics-agnostic pretraining on proxy task will be effective as long as the physics dataset (with PDE solution) is not dense. Under a rich PDE solution dataset setting, the performance gain from geometry-only pretraining may become minor.

Table 9: Comparison of prediction errors on the AirfRans dataset with scarce setting. Encoders are still pretrained on all 2400 geometry data, while only 200 sets of PDE solution labels are used to train neural operators.

| Dataset | Relative L2 on normalized data ($\times 10^{-2}$) | | | | | |
| --- | --- | --- | --- | --- | --- | --- |
| | GNOT | G+VAE | Transolver | T+VAE | LNO | L+VAE |
| AirfRans (Mesh) | 15.8 | 13.9 | 22.2 | 21.3 | 61.0 | 52.9 |
| AirfRans (Random) | 12.8 | 10.2 | 32.9 | 21.0 | 52.1 | 52.8 |

**Effect of physics data size.** We investigate the effectiveness of the latent geometry representation under different sizes of physics datasets. We expand the physics training data to 3800 for the Stress dataset ($\sigma_{vm}$ field on type 1 void) and the Electrostatics dataset ($\phi$ field on type 1 interface), while using the same VAE as in the main experiments (VAE3 in Table 3). Table 10 shows the performance of GNOT trained with different amount of physics data. For both datasets, we observe significant reduction in relative error for GNOT and GNOT+VAE when physics datasets are expanded. The performance improvement from replacing geometry representation is consistent across different sizes of physics data. Interestingly, the VAE used for the Stress dataset only observes 1900 random geometries generated from type 1 void prior, but still enhances the performance of GNOT when trained with 3800 physics data.

Table 10: Prediction error of GNOT (with and without physics-agnostic pretraining), trained with different number of physics data. We use the same VAE as in the main experiments.

| Dataset | Num of physics data | Relative L2 on normalized data ($\times 10^{-2}$) | | | |
| --- | --- | --- | --- | --- | --- |
| | | GNOT | | GNOT+VAE | |
| | Query method | Mesh | Random | Mesh | Random |
| | 950 | 18.4 | 17.6 | 15.4 | 12.9 |
| Stress | 1900 | 9.8 | 10.3 | 9.0 | 8.3 |
| | 3800 | 7.5 | 7.2 | 7.1 | 5.7 |
| | 900 | 6.4 | 6.7 | 5.1 | 5.3 |
| Electrostatics | 1800 | 4.2 | 4.6 | 3.3 | 3.4 |
| | 3800 | 3.1 | 3.3 | 2.3 | 2.3 |

**Masked Autoencoder.** We evaluate masked autoencoder (MAE) as an alternative pretraining method. The experimental results on the electrostatics dataset are shown in Table 11, where MAE(x) stands for masked autoencoder with mask ratio of x. We notice that MAE pretraining also achieve slightly improved accuracy compared to direct supervised learning. However, the improvement is rather marginal compared to our pretraining framework with occupancy proxy task. The physics-agnostic pretraining process via MAE attempts to predict the permittivity on some randomly masked points on the electrostatics dataset. This objective doesn't provide much information to the encoder, as there exists no prior constraints on the permittivity field $a$, unlike the solution field $u$. Also, MAE was designed for pixel space problems rather than point clouds. Therefore, we think MAE fits better for pixel space tasks or physics-aware pretraining such as learning representations of $u$ for downstream autoregressive task (predict $u_{t+1}$ from $u_t$).

Table 11: Comparison of prediction errors on the electrostatics dataset among direct supervised training on GNOT, our proposed pretraining strategy, and masked autoencoder pretraining.

| Dataset | Relative L2 on normalized data ($\times 10^{-2}$) | | | |
| --- | --- | --- | --- | --- |
| | GNOT | G+VAE | G+MAE(0.3) | G+MAE(0.7) |
| Electrostatics (Mesh) | 4.2 | 3.3 | 4.1 | 4.2 |
| Electrostatics (Random) | 4.6 | 3.4 | 4.3 | 4.3 |

**Effect of batch size.** For efficient training, a large batch size of 100 is employed in the main experiments (including error estimation) and ablation studies. Additionally, we assess the proposed framework's performance under a conventional batch size of 32. Table 12 presents the results of neural operators trained with identical settings as the main experiments 4, except for the smaller batch size. A reduction in relative error can be observed for all experiments when trained at a smaller batch size. Meanwhile, the latent geometry representation improves the performance of neural operator in most experiments. However, its impact is minimal in four of the comparisons: GNOT on Stress with mesh query, GNOT on Inductor (3D) with both queries, and Transolver on Inductor (3D) with mesh query. We also observe performance drop when training Transolver on the Stress dataset. A potential explanation is that simply replacing Transolver's first layer with a linear cross-attention layer may be insufficient to fully leverage the latent geometry representation.

Table 12: Comparison of neural operator performance with batch size of 32. All other model/training hyperparameters remain the same as in the main experiments. Experiments demonstrating significant performance enhancement are highlighted in bolded fonts.

| Dataset | Query | Relative L2 on normalized data ($\times 10^{-2}$) | | | | | |
| --- | --- | --- | --- | --- | --- | --- | --- |
| | | GNOT | G+VAE | Trans | T+VAE | LNO | L+VAE |
| Number of parameters | | 1.7-1.8 M | | 1.7-1.8 M | | 1.8-1.9 M | |
| Stress | Mesh | 6.8 | 6.7 | 6.9 | 7.6 | 13.1 | **9.3** |
| | Random | 6.1 | **5.5** | 6.2 | 6.7 | 14.4 | **6.5** |
| AirfRans | Mesh | 4.1 | **3.6** | 7.9 | **6.3** | 20.6 | **18.2** |
| | Random | 5.6 | **3.6** | 8.9 | **6.4** | 16.9 | **5.8** |
| Inductor (3D) | Mesh | 4.8 | 4.9 | 5.6 | 5.6 | 10.7 | **7.5** |
| | Random | 10.7 | 10.6 | 12.2 | **11.7** | 15.3 | **11.9** |
| Electrostatics | Mesh | 3.0 | **2.1** | 3.2 | **1.8** | 7.8 | **2.8** |
| | Random | 3.3 | **2.3** | 3.8 | **1.8** | 6.9 | **2.8** |

**Hidden dimension of neural operators.** Table 13 shows the results of neural operators with a hidden dimension of 256 and head number of 8, which drastically increases the number of learnable parameters. Larger neural operators achieve lower relative error compared to the main experiments in Table 2, with the learned geometry representation still outperforming point clouds in most cases. Minimal impact from the latent geometry representation is observed in two comparisons: GNOT on Stress with mesh query, and GNOT on Inductor (3D) with random query. Performance drop is observed in three cases: GNOT on Inductor (3D) with mesh query, Transolver on Stress with mesh query, and Transolver on Inductor (3D) with mesh query. Notably, this is the only scenario where

applying the latent geometry representation leads to performance degradation in GNOT. A potential explanation is that the iron core only takes a relatively small occupancy volume in 3D, making it difficult for the VAE (trained with a BCE reconstruction loss) to capture the geometry accurately. The performance drop in Transolver has been discussed in **Effect of batch size**.

Table 13: Comparison of neural operator performance with hidden dimension of 256. All other model/training hyperparameters remain the same as in the main experiments. Experiments demonstrating significant performance enhancement are highlighted in bolded fonts.

| Dataset | Query | Relative L2 on normalized data ($\times 10^{-2}$) | | | | | |
| --- | --- | --- | --- | --- | --- | --- | --- |
| | | GNOT | G+VAE | Trans | T+VAE | LNO | L+VAE |
| Number of parameters | | 6.9 M | | 6.9 M | | 7.6 M | |
| Stress | Mesh | 7.6 | 7.7 | 7.7 | 8.6 | 22.4 | **10.2** |
| | Random | 7.3 | **6.3** | 8.6 | **7.5** | 18.0 | **8.7** |
| AirfRans | Mesh | 6.0 | **4.1** | 9.2 | **8.5** | 25.3 | **13.9** |
| | Random | 6.1 | **4.2** | 12.0 | **8.1** | 19.4 | **9.4** |
| Inductor (3D) | Mesh | 5.4 | 6.1 | 6.7 | 7.2 | 21.5 | **8.2** |
| | Random | 11.4 | 11.3 | 13.4 | **12.0** | 15.9 | **12.1** |
| Electrostatics | Mesh | 3.4 | **2.5** | 4.3 | **2.2** | 10.2 | **3.1** |
| | Random | 3.8 | **2.6** | 4.6 | **2.2** | 9.8 | **3.1** |

**Size of latent tokens.** We ablate the VAE latent configuration in pretraining stage by comparing the performance gain on Stress dataset when attached to GNOT as shown in Table 14, where N*M in column headers indicates N tokens of dimension M. Results show that 256 latent tokens with 32 dimensions give the best performance across both mesh-based and random query strategies. Increasing token number or dimension doesn't yield consistent gains. Instead, the latent size appears to strike a balance between expressiveness and regularization.

Table 14: Comparison of prediction accuracy of GNOT on the Stress dataset using different sizes of pretrained latent. Columns denote the latent configuration (# token * latent dimension).

| Dataset | Relative L2 on normalized data ($\times 10^{-2}$) | | | | | | |
| --- | --- | --- | --- | --- | --- | --- | --- |
| | No VAE | 128*32 | 128*64 | 256*32 | 256*64 | 512*32 | 512*64 |
| Stress, Mesh | 9.8 | 10.0 | 9.6 | **9.0** | 9.5 | **9.0** | 9.2 |
| Stress, Random | 10.3 | 8.4 | **8.3** | **8.3** | 8.6 | 8.8 | 8.6 |

# E COMPUTE RESOURCES

We list the number of parameters and memory cost for training the VAEs and transformer-based neural operators in this section. Training time per epoch is not included as experiments are carried out on different GPUs (A100 or V100). To conduct experiments efficiently, we use a large batch size of 100 for all experiments. However, we also conduct extra experiments to examine the effectiveness of the proposed framework under a normal batch size of 32, as discussed in Appendix D. Other hyperparameters are listed in section 4.

Table 15: Number of parameters and memory consumption for physics-agnostic pretraining and operator learning at hidden dimension of 128. The difference among different experiments is negligible.

| Metric | VAE Encoder | VAE Decoder | GNOT | Transolver | LNO |
| --- | --- | --- | --- | --- | --- |
| Parameters (M) | 1.1 | 8.2 | 1.74 | 1.71 | 1.92 |
| Memory (GB) | 11.9 | | 29.9 | 32.1 | 7.5 |

## F    BROADER IMPACTS

This work introduces a physics-agnostic pretraining framework (generally adaptable to arbitrary mesh) to enhance the performance of neural operators. We expect positive impact in research and industry fields where partial differential equations are solved to simulate the physical world. However, we don't envision any misusing of our methods or causing any negative social impact.

