# OpenReview forum: "From Cheap Geometry to Expensive Physics: A Physics-agnostic Pretraining Framework for Neural Operators"
_ICLR.cc/2026/Conference — ICLR 2026 Poster_

### Official Review · Reviewer_pbnB · 2025-10-22

**Soundness:** 3
**Presentation:** 3
**Contribution:** 2
**Rating:** 4
**Confidence:** 4

**Summary:**

Pre-training of neural operators using geometry information.

**Strengths:**

- Well written. The author motivated the geometry-based pre-training strategy well.
- Considered various benchmark problems.
- Considered various recently proposed backbone architectures.
- Error bars.

**Weaknesses:**

- line 53: This method is also limited to varying geometry problems. To say something about applicability to general PDE problems, it is necessary to show effectiveness on a non-varying geometry problem.
- Relative $L^2$ errors are too high—for engineering applications, two to three digits of accuracy is usually recommended. However, more than 10% of error makes it difficult to see whether the improvement comes from the proposed method or optimization error.
- The paper tackled data scarcity. However, the reader may not agree that more than 800 training data points are scarce. Especially, the AirfRans dataset provides data data-scarce scenario. Reporting the result in such a scenario may boost the credibility of this work.
- Asserting the encoder as a learned chart for the moduli space is good, but there was no quantitative validation of it. Reporting the distance between two latent codes from two point clouds representing the same geometry would improve the claim.
- Perhaps, further truncation of the AirfRans domain was due to failure in the original domain. (In fact, the original domain of the AirfRans dataset had been truncated already.)

**Questions:**

- Training time ratio for 1st and 2nd stages?
- LNO is the most recent, but performed the worst. Could you comment on that?

---

> ### Author Response · Authors · 2025-11-20
>
> We thank the reviewer for the thoughtful and detailed feedback. Below, we respond to each point in turn, clarifying the scope of our contribution, providing additional quantitative evidence, and reporting new experiments where feasible.
>
> ***1. line 53: This method is also limited to varying geometry problems. To say something about applicability to general PDE problems, it is necessary to show effectiveness on a non-varying geometry problem.***
>
> The reviewer raises an important concern on the applicability of the proposed method on non-varying geometry problems. Our primary motivation is to relax the fixed grid/mesh requirement of prior neural operator pretraining frameworks, rather than to restrict ourselves to varying geometry.
>
> In operator learning, there are many possible input-output mappings, and we focus on learning a mapping from geometry to PDE solution as it is the most directly related to pretraining on varying grids (for this task, the grid can not be fixed). There are other operator learning tasks that can be defined on either fixed grids or varying grids, such as learning a mapping from an initial material distribution field to the electric potential field, or learning a mapping from SDF field to the stationary flow field. We also show the applicability of our method on these tasks (the electrostatic dataset, and alternative proxy task shown in Table 7), but we keep the varying grid setting which is more general and consistent with other experiments.
>
> **Conceptually, a method designed to handle varying grid/mesh structures is also applicable when the grid is fixed, as the latter is a special case.** In that setting, one would fix the coordinates of all points and choose an appropriate proxy input for pretraining (e.g., SDF or SV fields as in Table 7). A thorough empirical study on purely fixed grid benchmarks is an interesting extension but beyond the scope of this work.
>
>
> ***2. Relative L2 errors are too high. More than 10% of error makes it difficult to see whether the improvement comes from the proposed method or optimization error.***
>
> We thank the reviewer for raising this important question. As we have dicussed in section 4.2 of the manuscript, relative L2 error can appear very small without normalization when the mean value is huge and std is small. In such case, a model that consistently predicts the mean value also reaches low relative L2 error. Although most operator learning studies report the small relative L2 error on unnormalized datasets, we choose to report the normalized ones to show the true model performance. The table below gives an example of the L2 error after denormalization.
>
> ||GNOT (denormalized)|GNOT (normalized)|G+VAE (denormalized)|G+VAE(normalized)|
> |:---:|:---:|:---:|:---:|:---:|
> |Stress, Mesh ($\times 10^{-2}$)|0.16|9.8|0.15|9.0|
> |Stress, Random ($\times 10^{-2}$)|0.15|10.3|0.11|8.3|
>
> The stress dataset has mean value of 379156 and std value of 6180, making the relative L2 error almost two orders of magnitude smaller after denormalization. We believe that the model isn't actually achiecing over 99.8% accuracy as the std value is significantly smaller than the mean value. **Nevertheless, we still observe improvement on the unnormalized results.**
>
> ***3. The paper tackled data scarcity. However, 800 training data points not necessarily scarce. Especially, the AirfRans dataset provides data-scarce scenario. Reporting the result in such a scenario may boost the credibility of this work.***
>
> Thank you for this helpful suggestion. Indeed, AirfRans has a data-scarce setting where only 200 data are used for training. We test our framework under this data-scarce setting with the results shown in the table below.
>
> ||GNOT|G+VAE|Transolver|T+VAE|LNO|L+VAE|
> |:---:|:---:|:---:|:---:|:---:|:---:|:---:|
> |AirfRans, Mesh ($\times 10^{-2}$)|15.8|13.9|22.2|21.3|61.0|52.9|
> |AirfRans, Random ($\times 10^{-2}$)|12.8|10.2|32.9|21.0|52.1|52.8|
>
> Under such data-scarce setting, prediction accuracy gets reduced under all circumstances compared to the original setting as expected, and we can observe clear performance gain upon applying the two-stage training strategy in most experiments (except for LNO under random query). We believe that the stage 1 training on proxy task will be effective as long as the physics dataset (with PDE solution) is not dense. When there is a rich PDE solution dataset, the performance gain from stage 1 may become minor. We greatly appreciate this valuable suggestion, and we have included these results in the updated Appendix D.

---

> ### Author Response · Authors · 2025-11-20
>
> ***4. Quantitative validation of encoder (as a learned chart for moduli space). Reporting the distance between two latent codes from two point clouds representing the same geometry would improve the claim.***
>
> We greatly appreciate the reviewer for this insightful suggestion. Please refer to our updated Appendix B where we have included Figure 4 to visualize the distance between latent codes from point clouds representing the same and different geometries. Point clouds corresponding to the same geometry form tight, well-separated clusters, indicating that the encoder learns a meaningful representation in which distances in latent space correlate with differences between underlying geometries.
>
> ***5. Is further truncation of AirfRans due to failure in the original domain?***
>
> We thank the reviewer for this important question. The additional truncation of the AirfRANS domain is not due to a failure of the proposed method on the full domain, but for two reasons:
>
> - First, we hope to focus on the PDE solution near the airfoil. This is a strategy adopted in many operator learning studies such as [*A point-cloud deep learning framework for prediction of fluid flow fields on irregular
> geometries from Kashefi et al.*] and [*Geo-FNO from Li et al.*].
> - Second, it is an experiment design choice to avoid dataset specific tuning and keep the focus on the two-stage framework itself. In our main experiments, the stage 1 pretraining proxy task is an occupancy field, optimized with a BCE loss (Eq. 4). On the full AirfRANS domain, the airfoil occupies only about 1–2% of the computational domain, which makes this a highly imbalanced classification problem. A weighting strategy for the BCE loss would therefore be required to learn the occupancy field properly. To avoid introducing AirfRANS specific loss reweighting and hyperparameter tuning, we choose to work with a truncated domain where this imbalance is less severe.
>
> In Section 4.4 and Table 7, we also demonstrate alternative proxy tasks such as SDF and SV fields on AirfRANS. These alternative proxy tasks are implemented on the truncated domain for consistency, but they are directly applicable to the full domain setting without special treatment for class imbalance as they are regression tasks rather than classification. Due to time constraints in the rebuttal phase, we are not able to include additional full domain SDF or SV results, and we plan to explore this more systematically in future work.
>
> ***6. Training time ratio for 1st and 2nd stages?***
>
> The training time ratio for 1st and 2nd stages is approximately 4:1 for the stress dataset on GNOT (152 mins and 37 mins on single A100). However, the inference time is approximately 1:2 (details in our reponse 1 to reviewer En7x) as we only use the encoder in stage 1 for inference. This ratio can depend on model architectures, model sizes used in the two stages (for instance training LNO is faster than training GNOT), as well as the amount of unlabeled and labeled physical data. We have included training data information in Table 1 and the model size information in Appendix E.
>
> However we didn't optimize the training pipelines for best utilization of GPU memory and bandwith as the focus of this work is not on computation efficiency.
>
> ***7. LNO is the most recent, but performed the worst. Could you comment on that?***
>
> The reviewer points out this interesting phenomenon that LNO performs worst though published more recently. We believe that each neural operator architecture has its own advantage. For LNO, its biggest advantage is the high computation and memory efficiency. The performance of each neural operator model can also vary across different datasets and tasks. Although LNO doesn't show best performance on the tasks in this paper, it might achieve higher accuracy on fixed grid problems or inverse problems as discussed in the LNO paper.
>
> We also want to make a note that the purpose of this work is not to compare among latest neural operator architectures. Therefore, we avoid extensive hyperparameter tuning and keep model size and architectures consistent across experiments to focus on the performance of the proposed two-stage training framework.

---

> > ### Comment · Reviewer_pbnB · 2025-11-20
> >
> > Thank you very much for the detailed replies.
> >
> > **1. and 5.**
> >
> > > Our primary motivation is to relax the fixed grid/mesh requirement of prior neural operator pretraining frameworks, rather than to restrict ourselves to varying geometry.
> >
> > Thank you for the clarification. In that case, I believe comparisons with existing pre-training approaches on the electrostatic dataset are necessary, since (1) the main goal of this paper is to propose a pre-training method, and (2) the dataset has fixed geometry (only the permittivity varies).
> >
> > > The additional truncation of the AirfRANS domain is not due to a failure of the proposed method on the full domain, but for two reasons:
> >
> > I could not find statements in the cited works indicating that they truncated the domain for the airfoil design problem. Typically, the computational domain would already be chosen to capture the physically relevant quantities for the design task. Moreover, I believe the second reason provided for truncation weakens the authors’ claim that the proposed pre-training approach can, in principle, be applied to fixed-geometry problems.
> >
> > **2. and 6.**
> >
> > Given that operator networks aim to approximate a function-to-function mapping, using a *denormalized* relative $L^2$ error---corresponding to a distance in function space---would be more natural. However, since the percentage improvements are similar, I do not mind to using the normalized $L^2$ error. Thank you for the additional table. I leave it to the reader to judge whether pre-training that is four times more expensive than NO training is worthwhile for an improvement from 0.0015 to 0.0011.
> >
> > **4.**
> >
> > > Please refer to our updated Appendix B where we have included Figure 4 to visualize the distance between latent codes from point clouds representing the same and different geometries. Point clouds corresponding to the same geometry form tight, well-separated clusters.
> >
> > I think it is necessary to visualize the actual geometries within the same cluster and from different clusters, as t-SNE and UMAP can sometimes be misleading [Stop Misusing t-SNE and UMAP for Visual Analytics, Jeon et al., 2025].
> >
> > **3. and 7.**
> >
> > I sincerely appreciate the authors’ responses.

---

> > > ### Author Response · Authors · 2025-11-26
> > >
> > > Thank you for the insightful comments. Please find our response below.
> > >
> > > ***Response 1 and 5:***
> > >
> > > **Compare with existing pretraining methods**
> > >
> > > As discussed in the manuscript, most existing pretraining methods for operator learning are physics-aware and therefore not suitable for our motivation. The only remaining option would be masked autoencoder as discussed in *[Data-Efficient Operator Learning via Unsupervised Pretraining and In-Context Learning by Chen et al.]*. We have adapted its implementation to the electrostatic dataset with the results shown below, where MAE(x) stands for masked autoencoder with mask ratio of x, pretrained on the geometry only data.
> > > ||GNOT|G+VAE|G+MAE(0.3)|G+MAE(0.7)|
> > > |:---:|:---:|:---:|:---:|:---:|
> > > |Elec, Mesh ($\times 10^{-2}$)|4.2|3.3|4.1|4.2|
> > > |Elec, Random ($\times 10^{-2}$)|4.6|3.4|4.3|4.3|
> > >
> > > We notice that MAE pretraining also achieve slightly improved accuracy compared to direct supervised learning. However, the improvement is rather marginal compared to our pretraining framework with occupancy proxy task (we achieve more significant improvement). By intuition, the pretraining objective of MAE on the electrostatic dataset would be to predict the permittivity on some randomly masked points which doesn't provide much information to the encoder, as there exists no prior constraints on the permittivity field $a$, unlike the solution field $u$. Also, MAE was designed for pixel space problems rather than point clouds. That said, we think MAE fits better for pixel space tasks or physics-aware pretraining such as learning a good representation of $u$ for downsteam autoregressive task (predict $u_{t+1}$ from $u_t$).
> > >
> > > We have also updated Appendix D to include these results.
> > >
> > > **Domain truncation for AirfRans**
> > >
> > > - The two cited works don't explicitly mention domain truncation. The first citation mentioned that the neural network domain is only a subdomain of the CFD domain in the data generation section. The second citation emphasized the deformed mesh near the airfoil, which can be seen in Fig. 8 where the domain boundary is very close to the airfoil.
> > > - The second reason: avoid problem specific weighting strategy for BCE loss, is a limitation for tiny geometry rather than fixed geometry. For problems with tiny geometry, one needs the weighting strategy for stage 1 pretraining as only a tiny portion of query points show positive occupancy in BCE loss. But this is not a concern for problems with fixed geometries.
> > >
> > > ***Response 2 and 6:***
> > >
> > > Thank you for the feedback. We acknowledge the increased training cost for stage 1 pretraining. However, the core motivation of this work is to mitigate the cost in data collection rather than training cost, which is minor in scientific domains. In industrial applications, the expense of solving PDEs and obtain physical field labels is too large. We have included the computation costs for solving PDEs and mere mesh generation in the table below, where $D$ have physical field labels $u$, while $D'$ only have geometry information $a$:
> > >
> > > ||Stress|AirfRans|Inductor|Electrostatics|
> > > |:---:|:---:|:---:|:---:|:---:|
> > > |Data generation time for $D$ (CPU hours)|270|7680|1200|130|
> > > |Mesh generation time for $D'$ (CPU hours)|<0.1|0.14|<0.1|<0.1|
> > >
> > > In most operator learning scenarios, the majority of the computation cost comes from PDE solution generation. The extra 2-3 hours of pretraining is negligible compared to the computation cost of generating labeled simulation data. As discussed in the manuscript, the high cost of training label generation is precisely what motivates our physics-agnostic pretraining framework.
> > >
> > > We have updated Table 1 of the manuscript to reflect the computation cost for solving PDEs and mere mesh generation.
> > >
> > > ***Response 4:***
> > >
> > > We have updated the manuscript and included visualization of the actual geometries in Figure 4. Besides the observation that point clouds from the same geometry form tight clusters, we also see that geometries with similar high-level features lie close to each other in the latent space: for example, geometries 1 and 3 have holes elongated in the horizontal direction, while geometries 2 and 4 have holes elongated in the vertical direction.

---

> > > > ### Comment · Reviewer_pbnB · 2025-11-26
> > > >
> > > > Thank you very much for answering my questions. I have raised my score to 6.
> > > >
> > > > > The extra 2-3 hours of pretraining is negligible compared to the computation cost of generating labeled simulation data.
> > > >
> > > > Perhaps, showing that G+VAE requires less PDE solution data to achieve the same accuracy as GNOT would be a more effective demonstration.
> > > >
> > > > > The second reason: avoid problem specific weighting strategy for BCE loss, is a limitation for tiny geometry rather than fixed geometry.
> > > >
> > > > Let me put that differently. As you’ve described, the problem with tiny airfoil is that most query points will be outside the airfoil. I believe this is essentially the same as fixed geometry, where we don’t consider the airfoil in the domain. Please correct me if I’m mistaken.

---

> > > > > ### Author Response · Authors · 2025-11-27
> > > > >
> > > > > Thank you for the positive assessment of our work.
> > > > >
> > > > > ***Perhaps, showing that G+VAE requires less PDE solution data to achieve the same accuracy as GNOT would be a more effective demonstration.***
> > > > >
> > > > > Indeed, showing the performance of GNOT and G+VAE at different amount of PDE solution data can be a more effective demonstration. We had an experiment on number of physics data on the stress and electrostatics dataset in Appendix D Table 10. We will consider replacing it with a plot for better visualization.
> > > > >
> > > > > ***As you’ve described, the problem with tiny airfoil is that most query points will be outside the airfoil. I believe this is essentially the same as fixed geometry, where we don’t consider the airfoil in the domain.***
> > > > >
> > > > > Thank you for the clarification and we would like to refine our explanation. In the AirfRans setup, the computation domain indeed doesn't include the interior part of airfoil (PDE solution $u$ learned in stage 2 is defined outside of airfoil). However, the occupancy proxy task in stage 1 pretraining is defined over the entire bounding box, including the interior of airfoil. As shown in Figure 2, there are query points (in stage 1) both inside and outside the airfoil. And when airfoil is tiny, the number of blue points (inside) is much smaller than the number of red points (outside), leading to heavily imbalanced BCE loss. While in stage 2, there are either no query points inside the airfoil or predictions at such points are masked out. Therefore, we think the limitation is tied to tiny occupancy regions rather than fixed geometry.
> > > > >
> > > > > In a truly fixed geometry setting for AirfRans, the only varying inputs would be two global parameters: wind speed and attack angle. In that case, our framework would be less beneficial since it's designed to extract physics-agnostic field level information (geometry or other input fields) in stage 1. When the only inputs are global scalars, there is little structure for the encoder to exploit.
> > > > >
> > > > > We hope this clarifies that (1) the BCE weighting issue is about tiny geometric support in the occupancy task, and (2) our method is most useful when the input contains nontrivial spatial structure (varying geometry or fields), rather than when the geometry is fixed and only global parameters vary. Please let me know if there is misunderstanding or you have further questions.

---

### Official Review · Reviewer_hYfi · 2025-10-28

**Soundness:** 4
**Presentation:** 4
**Contribution:** 4
**Rating:** 8
**Confidence:** 4

**Summary:**

The paper proposes a two-stage training framework for neural operator learning that leverages geometry only data (like cheap, unlabeled, physics-agnostic data) to improve performance on PDE solution predictions (via expensive, label-scarce physics data). This approach combines representation learning and operator learning, allowing physics-informed models to benefit from geometry-only datasets. This is typically ignored in standard PDE modeling.

**Strengths:**

* well-motivated framework bridging geometric pretraining and operator learning. PDE-based simulations (like CFD, elasticity, electromagnetics and so on) are computationally expensive, while 3D geometries are normally cheap to obtain.
* The occupancy field proxy is neat. It’s simple, generalizable and physics-agnostic. It seems to effectively force the encoder to learn spatial and structural priors that are transferrable to PDE learning tasks.
* The authors test some alternative proxy tasks (like signed distance fields, shortest vectors etc) and show comparable or improved results, especially for CFD datasets.
 * There is nice integration with Transformer-based Neural Operators. The latent embeddings are seamlessly fed into three state-of-the-art transformer-based neural operators. This makes for broad applicability and makes the contribution compatible with ongoing work in neural operator transformers.
* Demonstrated generality across problems and architectures
* Good experiments
* The authors are honest about limitations

**Weaknesses:**

* Lack of joint optimization between pretraining and PDE stages.
* Limited exploration of theoretical connections between learned latent and PDE operator manifolds.
* Real-world benchmarks (e.g., turbulence, multiphysics coupling and stuff like that) could further validate scalability. The latter is important yet not discussed enough.
* The authors themselves allude to some limitations / weaknesses. Complex geometries with multiple interacting components may require multi-class or more expressive proxies. The lack of joint fine-tuning may limit performance.

**Questions:**

On page 1 you comment on fixed-grid structures. Don’t you mean fixed topology?

Pg2 input function space is assumed to contain only geometry info. How much of a limitation do you see this as, and how might this be extended?

Pg4 you define \epsilon \sim N(0,1). Then on pg 5 you have \epsilon as the scale param in another N(0,\epsilon I). The latter use implies \epsilon > 0, at odds with the first usage definition.

Pg6 you exclude hyperparam tuning. I get your focus on the 2-stage training – but surely one of the selling points of the work is the superior performance, so it’s important to understand better the context of the reported results. Can you comment more?

Pg 7 – re. comment about unnorm data sets and MSE. Can’t you present NMSE, or R^2?

References – you should protect capitals – like {N}avier-{S}tokes etc.

---

> ### Author Response · Authors · 2025-11-20
>
> We greatly appreciate the reviewer for the detailed and insightful feedback. Please find below our response to each points.
>
> ***1. Lack of joint optimization between pretraining and PDE stages.***
>
> Thank you for pointing out this important direction. Since the data in $D$ and $D'$ (section 2.1 in manuscript) are not fully aligned, it is natural to consider leveraging two stages for separate optimization. In the current work, we provide only a preliminary validation for the proxy task pretraining framework.
>
> We agree that training the two stages separately may leave room for performance improvement. In our current framework, we freeze the pretrained encoder during stage 2 to prevent the limited PDE data from overwriting the general geometric representations learned in stage 1. This design choice prioritizes stability and data efficiency, which are crucial for the scarce PDE label scenario we focus on.
> However, we fully acknowledge the value of joint optimization. As a future extension, we plan to explore three promising directions:
> - Sequential finetuning with a small learning rate to allow the encoder to adapt its geometric representations to specific PDEs.
> - Hybrid loss functions that couple occupancy reconstruction (stage 1) and PDE prediction (stage 2) for end-to-end optimization.
> - Insertion of adapter-style modules into the encoder so that general geometric knowledge is preserved while allowing controlled PDE-specific adjustments.
>
>
> ***2. Limited exploration of theoretical connections between learned latent and PDE operator manifolds.***
>
> We thank the reviewer for this insightful comment. We agree that understanding the theoretical connection between the learned latent space and the underlying PDE operator manifold is both important and intellectually compelling.
>
> Our work lays preliminary foundations for such a theoretical perspective. As discussed in Appendix B, we interpret stage 1 pretraining as learning a coordinate chart on the moduli space of shapes, where the encoder E maps raw point clouds into a structured latent manifold. We have also included a t-SNE visualization for the latent space in Appendix B as an empirical support.
>
> To further address the reviewer’s question, we have developed a theoretical framework that decomposes the generalization error of our two-stage method. The central result is the following error decomposition theorem:
>
> **Theorem (Error Decomposition).** The generalization error of our two-stage operator learning framework can be bounded by three terms:
> - Representation Error: $O(C_E/\sqrt{N_{geo}})$. This term reflects how well the stage 1 pretraining captures geometric features that are relevant to the underlying PDE physics. The constant $C_E$ is small if the latent space provides a "smoother" or "more structured" input domain for the PDE operator.
> - Estimation Error: $R_N(F_θ)+O(1/\sqrt{N_{phy}})$. This is the standard error from learning the neural operator in stage 2, given the finite physics data.
> - Approximation Error: $ϵ_{approx}$​. This is the inherent error from using a specific neural operator architecture.
>
> **This theorem provides a rigorous explanation for our empirical success:**
> - Our key contribution is the introduction of the pretraining stage, which directly minimizes the representation error term. By leveraging abundant unlabeled geometries ($N_{geo}\gg N_{phy}$), we effectively reduce the dominant source of error in data-scarce regimes.
> - The constant $C_E$​ formalizes the 'theoretical connection' mentioned by the reviewer. A good proxy task (like occupancy) ensures that $C_E$​ is small, so that the latent space is well-aligned with the PDE operator manifold. Our ablation studies on alternative proxy tasks (SDF, SV in Table 7) empirically validate this: different proxies lead to different $C_E$​, hence different final performance.
> - The theorem explains the phenomenon of "consistently improved prediction accuracy". Even with a fixed $N_{phy}$ (the scarce physics data)​, increasing $N_{geo}$​ (the abundant geometry data) in stage 1 tightens the overall error bound. This is precisely what we observe in our ablation study on the number of geometry data (Table 4), where performance improves as more geometry data is used for pretraining (VAE1 → VAE3).
>
> We will refine and include this theoretical analysis and its interpretation in the final version of the paper.

---

> > ### Comment · Reviewer_hYfi · 2025-11-25
> >
> > I appreciate your responses to the points raised in my review.
> > Regarding the lack of joint optimization, you mention
> >   | As a future extension, we plan to explore three promising directions:
> > is this planned for generic future work, or do you plan to do [some of] this for a revised version of this paper?

---

> > > ### Author Response · Authors · 2025-11-27
> > >
> > > We consider these three directions as generic future work, as they require very different training frameworks with (potentially) a lot of tuning work, and it's difficult to include these results.

---

> ### Author Response · Authors · 2025-11-20
>
> ***3. Real-world benchmarks (e.g., turbulence, multiphysics coupling and stuff like that) could further validate scalability.***
>
> Thank you for these suggestions regarding more realistic benchmarks and scalability.
>
> - **Turbulence benchmarks:** Please refer to our response 4 to reviewer En7x.
> - **Multiphysics benchmarks:** Most existing operator learning benchmarks focus on single physics problems (may include more than one physical field variable). We only predict single channel physical field in this work for simplicity, but we can easily extend to multiple physical field variables by adding more output channels in stage 2. On the other hand, multiphysics datasets (coupling among multiple PDEs) are still relatively limited in the literature. While there are emerging efforts in this direction (e.g., Multiphysics Bench from Yang et al., Rayleigh-Benard Convection in CoDA from Rahman et al.), evaluation on such datasets is beyond the scope of the present paper. But we agree that applying our approach to multiphysics coupling scenarios would be an interesting direction for follow up work.
> - **Scalability:** In this work, we focus on problems with moderate mesh sizes to provide a preliminary validation of the proposed two-stage framework. Evaluating our framework on large scale datasets such as DriverNet++ is indeed valuable but requires substantial engineering and computational resources. We therefore leave this to future studies.
>
> ***4. The authors allude to some limitations / weaknesses. Complex geometries with multiple interacting components may require multi-class or more expressive proxies. The lack of joint fine-tuning may limit performance.***
>
> Thank you for highlighting these limitations. We share your perspective and appreciate the opportunity to elaborate on potential mitigation strategies.
>
> For complex geometries involving multiple interacting components (e.g., assembled mechanical systems or multi-body fluid domains), we agree that the binary occupancy proxy is not expressive enough, as it cannot differentiate between distinct components. As noted in the paper, we plan to extend the proxy task to multi-class occupancy, where each component receives an independent label and the model is trained using a cross-entropy objective. As the proposed framework is flexible in proxy task selection, additional proxies such as signed distance fields or hybrid geometric descriptors can be incorporated in stage 1 to ensure the encoder captures richer geometric information relevant for downstream PDE learning. This is an important direction, as geometry alone may not fully encode all the structure needed for PDE prediction in multi-component scenarios.
>
> Regarding the lack of joint finetuning, please refer to Response 1.
>
> ***5. On page 1 you comment on fixed-grid structures. Don’t you mean fixed topology?***
>
> Yes, the reviewer is correct. This is a word ambiguity, and we appreciate you catching it. We have updated the manuscript accordingly.
>
> ***6. Pg2 input function space is assumed to contain only geometry info. How much of a limitation do you see this as, and how might this be extended?***
>
> Thank you for raising this important point. The assumption that the input function space in Pg2 contains only geometric information is indeed a simplification for controlled validation. We acknowledge that this becomes a limitation when dealing with more general PDE settings in which boundary conditions (BC), source terms, or material properties vary across samples. We believe that the current architecture can be naturally extended in the following ways to incorporate additional input information:
> - **Alternative proxy task during Stage 1 pretraining:** The occupancy field proxy reconstruction task in stage 1 pretraining can be replaced with regression for specific input fields (e.g., BC values, material parameters, source terms). In fact, the manuscript already includes a small experiment for leveraging signed distance fields or shortest vector fields as stage 1 proxy task (section 4.4 and Table 7). This can be easily extended to other input field proxies.
> - **Concatenation or multi-branch encoder for multiple input fields:** When multiple input fields exist, they can be concatenated to allow the VAE to learn joint latent representations that integrate both geometry and physically relevant metadata. Alternatively, we can have multiple encoder branches each dedicated to a specific input function. Cross-attention or feature-fusion modules can then integrate the multiple streams.
>
> These extensions do not require modifying the overall two-stage philosophy. They simply broaden the input modality of stage 1. We will clarify this in the revision and include a discussion of how the framework can be generalized to richer PDE inputs.

---

> ### Author Response · Authors · 2025-11-20
>
> ***7. Pg4 you define \epsilon \sim N(0,1). Then on pg 5 you have \epsilon as the scale param in another N(0,\epsilon I). The latter use implies \epsilon > 0, at odds with the first usage definition.***
>
> Thank you for pointing out this notation inconsistency. We have replaced the $\epsilon$ symbol on pg5 with $\zeta$, which appears as a small positive scalar controlling the variance of the perturbation distribution $N(0,\zeta I)$ used to create noisy query points.
>
> ***8. Pg6 you exclude hyperparam tuning. I get your focus on the 2-stage training – but surely one of the selling points of the work is the superior performance, so it’s important to understand better the context of the reported results. Can you comment more?***
>
> We appreciate the reviewer's attention to the experimental setup. What we mean on pg6 is that we keep most hyperparameters (embedding dimension, MLP factor, activation, learning rate, batch size, scheduler, etc.) consistent across all experiments to ensure a fair and controlled comparison focused on the effectiveness of the two-stage training strategy itself, rather than finding a specific set of hyperparameter to improve the absolute performance of any particular model. We only change the layer number of neural operators to ensure they share approximately the same number of trainable parameters. That said, we did perform reasonable baseline tuning for this set of global hyperparameters. However, we keep all hyperparameters consistent when comparing our two-stage settings with standard single-stage settings, and we observe consistent and statistically significant improvements across multiple architectures and datasets.
>
> We agree that further tuning could push absolute performance higher, and there is certainly improvement room for LNO if we assign a specific set of hyperparameters. Nevertheless our goal is to demonstrate that the pretrained geometry representation provides a systematic boost, regardless of the specific operator architecture.
>
> ***9. Pg 7 – re. comment about unnorm data sets and MSE. Can’t you present NMSE, or R^2?***
>
> Thank you for the suggestion. Relative L2 error (Eq. 5) is a widely adopted and robust metric for operator learning tasks where the prediction is a field. Numerically, this metric is equivalent to NMSE up to a square root, so the information conveyed is essentially the same. Without data normalization, either rel L2 or NMSE may fail to report the true model performance when the ground truth has a large mean value. Please refer to our reponse 2 to reviewer pbnB where we present the rel L2 on denormalized stress fields (significantly smaller compared to normalized stress fields).
>
> Regarding $R^2$, it is definitely a good performance evaluation metric (for general regression tasks) regardless of data normalization as it reports the proportion of predictable variation. However, there is no universally accepted definition of $R^2$ for field prediction tasks as the number of query points as well as their locations vary from data to data. Therefore, choices such as per-point $R^2$, global $R^2$, or variance-normalized function norms can produce inconsistent interpretations.
>
> ***10. References – you should protect capitals – like {N}avier-{S}tokes etc.***
>
> We thank the reviewer for bringing this to our attention. We have updated the manuscript accordingly.

---

### Official Review · Reviewer_xrAn · 2025-10-31

**Soundness:** 3
**Presentation:** 2
**Contribution:** 1
**Rating:** 2
**Confidence:** 4

**Summary:**

The authors of this work proposed a novel deep learning framework for the prediction of quantities of interest in computational mechanics such as stress, pressure, and velocity, where the solution of PDEs depend on the geometry of the computational domain. The tested their network by choosing dataset from fluid, solid, and electrical engineering and showed the performance by reporting relative errors in L2 norm. The study includes both 2D and 3D examples. At high level, the network has two stages, in the first stage, there is an encoder that only learns the geometric feature of computational domain from a set of point clouds. In stage 2, there is operator learning to learn PDE variables. Combing the features learned from stage 1 and stage 2, the network predicts variables in query coordinates.

**Strengths:**

The article is about an important subject in computational mechanics where we would like to accelerate geometric design, depending on various parameters, and we use deep learning frameworks for this goal. So, the topic is important. The authors investigated the performance using different datasets both from fluid and solid mechanics as well as electrical engineering (rather than only one dataset or one specific application).

**Weaknesses:**

As I mentioned in the Strengths section, the problem has been well defined. But I strongly believe that to overcome the challenge of this problem, there have been a class of point-cloud based neural networks (e.g., PointNet, PointNet++, etc.) that have been used for computational mechanics problems. PointNet has been introduced in 2017 and the usage of PointNet in CFD was introduced in 2021 for the first time and later on, there have been so many usage of PointNet in this area. Here is a list of some papers:

PointNet and PointNet ++ in computer graphics:

https://openaccess.thecvf.com/content_cvpr_2017/html/Qi_PointNet_Deep_Learning_CVPR_2017_paper.html

https://proceedings.neurips.cc/paper/2017/hash/d8bf84be3800d12f74d8b05e9b89836f-Abstract.html

PointNet for CFD:

https://doi.org/10.1063/5.0033376

PointNet for PINNs:

https://doi.org/10.1016/j.jcp.2022.111510

PointNet for Operator Learning:

https://doi.org/10.1016/j.cma.2024.117130

PointNet for KANs:

https://doi.org/10.1016/j.cma.2025.117888

None of these papers have been cited by the authors. So, I imagine that they have been not aware of these methodology. Because these networks handle the same issue mentioned by the authors in more efficient ways. There is only one network that learns both geometric features and also PDE variables (e.g., velocity, pressure, temperature, etc.) simultaneously. The proposed method in this article, while works, but it is very complicated in comparison. It has two stages and requires pre and post processing as explained by the authors. In simple words, the authors tried to learn geometry and variables in two different networks, while both can be learned simultaneously in one network. I like the idea; however, we need to introduce something more efficient compared to what have been developed so far; otherwise, what is the motivation for using this new method.

A few other comments:

I think Figure 1 is redundant, Figure 2 can be also presented better, in the main body of the article, we could have more visual results and comparison between different methods. There are some bold words in abstract, I do not think it is good to make them bold. Introduction is relatively long for ICLR and the methodology has been explained very briefly. Moreover, in the results, again there are some details about the dataset which could be moved to the supplementary sections. Numbers in tables are written using a large, unusual font sizes.

**Questions:**

Have you ever tried PointNet, PointNet++, and other point-cloud based neural networks for solving these problems? Please see the details of my explanation in the Weaknesses section.

---

> ### Author Response · Authors · 2025-11-20
>
> Thank you for the feedback. We would like to first clarify that **this work lies in the domain of operator learning tasks rather than for point cloud classification/segmentation tasks. Also, our intention is not to propose or compare any model architecture. Instead, we are proposing a pretraining framework to effectively extract information from a large physics-agnostic dataset and empower the performance of neural operator on physical field prediction.** This is motivated by an interesting and common scenario in industry where engineers have plenty of candidate geometry designs but lack computation resource to get PDE solutions for all designs.
>
> The reviewer mentioned that PointNet can also predict physical fields from point clouds and listed six literatures, among which two journal papers are related to operator learning: [*PointNet for CFD*] and [*PointNet for Operator Learning*]. However, these two literatures focus on addressing specific engineering problems rather than establishing model architecture for general operator learning tasks. They don't provide comprehensive baseline comparisons across multiple datasets nor the code implementation.
>
> We acknowledge that PointNet/PointNet++ can be used for operator learning tasks. But we didn't include them as baselines for three reasons:
> - PointNet and PointNet++ were proposed 8 years ago for classification and segmentation tasks and were not optimized for operator learning tasks. As a result, their performance (accuracy and memory efficiency) is not comparable to the recent neural operator models which are readily included in our work.
> - It is not a common practice to include PointNet/PointNet++ as baselines in operator learning research.
> - We are proposing a pretraining framework for operator learning task rather than any model architecture.
>
> We add a brief introduction to PointNet in the related work section, and conduct a small experiment on PointNet++. More details are included in our reponse to each point as below.

---

> ### Author Response · Authors · 2025-11-20
>
> ***1. Have you ever tried PointNet, PointNet++, and other point-cloud based neural networks for solving these problems?***
>
> We thank the reviewer for raising this concern. As discussed above, PointNet is usually not considered as a baseline model for operator learning tasks. But we have included a small comparison for PointNet++ as shown in the table below.
>
> ||GNOT|G+VAE|Transolver|T+VAE|PointNet++|
> |:---:|:---:|:---:|:---:|:---:|:---:|
> |Stress, Mesh ($\times 10^{-2}$)|9.8|9.0|11.5|11.2|13.4|
> |Stress, Random ($\times 10^{-2}$)|10.3|8.3|11.5|9.7|13.7|
> |Trainable Parameters (M)|1.74|1.74|1.71|1.71|1.85|
> |Memory (GB)|29.9|29.9|32.1|32.1|73.6|
>
> PointNet and PointNet++ are indeed famous models for point cloud classification and segmentation tasks. However, they were proposed years ago with a different focus from operator learning. Therefore, PointNet++ shows larger prediction error, lower memory efficiency, with approximately doubled training time (76 minutes on single A100) compared to GNOT (37 minutes on single A100). Unfortunately, our two-stage training framework is not directly adaptable to PointNet++ as we use tokenized latent space in stage 1.
>
> ***2. The authors tried to learn geometry and variables in two different networks, while both can be learned simultaneously in one network.***
>
> Thank you for pointing out this valuable question. We use two different models as we have two training tasks to better exploit two qualitatively different datasets:
>
> $$
> D= \lbrace (a_k,u_k) \rbrace ^ {|D|} _ {k=1}, \quad D'= \lbrace (a_j) \rbrace ^ {|D'|} _ {j=1}, \quad |D'| \gg |D|
> $$
> - $a$: geometry point cloud
> - $u$: **expensive** PDE solution
>
> Existing operator learning literatures only focus on supervised learning on $D$ without providing a proper framework for leveraging $D'$, especially for irregular mesh. We address this limitation by training a VAE using geometry information $a$ from $D\cup D'$ via a proxy reconstruction task $o$, and integrate the VAE encoder with the neural operator in stage 2 training for better prediction accuracy on $u$. Notice that the two models share the same input $a$ but have different output.
>
> We agree that there exists possibility of performing stage 1 pretraining and stage 2 operator learning simultaneously in one network on both $D$ and $D'$. However, this is not realized in any existing operator learning literature. And we think this is an interesting future research direction.
>
> ***3. Figure 1 is redundant. Bold words are not good. Introduction is long. Some results can be moved to supplementary. Numbers in table have large font.***
>
> Thank you for raising these concerns. We want to clarify that Figure 1 is intended to make the scope of our work explicit: we focus on a physics-agnostic pretraining framework for operator learning tasks, rather than proposing a new encoder-decoder neural operator architecture.
>
> Regarding the other remarks, we have removed the bold words in abstract, but we note that the paper adheres to the official ICLR style and formatting guidelines. We also believe that there is some unavoidable subjectivity in preferences about structure, emphasis, and writing sytles. These aspects do not affect the technical soundness or novelty of the work. That said, we are happy to make minor adjustments in a camera-ready version to better align with the general styles, while keeping the core structure and message intact.

---

> ### Author Response · Authors · 2025-11-28
>
> As the rebuttal deadline is approaching, we would be very grateful if reviewer xrAn could let us know whether our clarifications adequately address your concerns.

---

### Official Review · Reviewer_En7x · 2025-11-03

**Soundness:** 2
**Presentation:** 3
**Contribution:** 3
**Rating:** 4
**Confidence:** 5

**Summary:**

This paper introduces a two-stage operator learning framework that first learns a latent geometric representation using an occupancy prediction model, followed by conditional inference for query locations through a transformer-based neural operator. The method is validated on four 2D and 3D examples by comparing relative L2 errors across three major operator learning models. The results consistently demonstrate performance improvements over the baseline models. The paper also discusses how increased dataset variability enhances performance and analyzes the effect of different KL coefficients.

**Strengths:**

1. The idea presented in this paper is clear and well-motivated. The use of cross-attention for learning latent geometric representations is interesting and may have broader applicability to PDE-based surrogate modeling approaches.
2. Table 2 provides a comprehensive comparison between the proposed method and three other approaches, enhancing the credibility of the results.
3. The ablation studies are clearly presented and address two important aspects: the effect of KL weights and the impact of geometric data variability.

**Weaknesses:**

1. The paper mainly evaluates model performance in terms of relative L2 error. However, introducing Stage 1 may introduce additional computational costs, which are not measured. Moreover, the advantages of transforming the original point cloud into a latent geometric representation are not fully explored. For instance, could this help reduce inference time for temporal propagation since the propagation occurs in latent space?
2. Table 6 shows that an AE alone suffices for the AirfRANS dataset, making VAE unnecessary. A similar analysis on the other datasets would help clarify this observation.
3. Finally, there is no comparison with other geometry-agnostic methods, such as “Li, Zongyi, et al. Geometry-informed neural operator for large-scale 3D PDEs. Advances in Neural Information Processing Systems 36 (2023): 35836–35854.” Without such comparisons, it is difficult to assess the effectiveness of the proposed method fully.

**Questions:**

1. A clear computational cost study is needed to show a balance of the performance win and the speed reduction since the proposed approach introduces an additional stage compared to other models.
2. How to address problems with sharp boundaries and more chaotic dynamics, such as turbulence, with the proposed approach?

---

> ### Author Response · Authors · 2025-11-20
>
> We would like to thank the reviewer for their helpful suggestions. We hope to first clarify that the **focus of our work is not on model achitectural improvement. Instead, we propose a physics-agnostic pretraining framework (for irregular point clouds) to better exploit two qualitatively different datasets:**
>
> $$
> D= \lbrace (a_k,u_k) \rbrace ^ {|D|} _ {k=1}, \quad D'= \lbrace (a_j) \rbrace ^ {|D'|} _ {j=1}, \quad |D'| \gg |D|
> $$
> where **expanding** $D$ **is difficult, but expanding** $D'$ **is cheap.**
> - $a$: geometry point cloud
> - $u$: **expensive** PDE solution
> - Stage 1 pretraining: $(a,o)$ pairs from $D\cup D'$ where $o$ is obtained from a cheap physics agnostic proxy task
> - Stage 2 training: $(a,u)$ pairs from $D$
>
> This is an interesting and common scenario in industry where engineers have plenty of candidate geometry designs but lack computation resource to get PDE solutions for all designs. Therefore, the purpose of stage 1 pretraining and construction of the latent space is to extract geometry knowledge from the rich and physics-agnostic dataset $D'$ to enhance stage 2 operator learning on $D$.
>
> We summarize and clarify the questions and concerns below:
>
> ***1. Computation costs for two-stage inference and potential efficiency gain on temporal problems via latent space propagation.***
>
> Thank you for raising this important point. The inference time for a single data on Nvidia A100 is shown in the table below:
> ||VAE Encoder|GNOT|Transolver|LNO|
> |:---:|:---:|:---:|:---:|:---:|
> |Time (ms)|2.0|5.5|4.3|4.0|
>
> The size of each model is listed in Appendix E. Notice that although the entire size of VAE is larger than neural operators, the encoder only takes a small portion of parameters (decoder takes the most trainable parameters). Therefore, the additional computation cost at inferece is relatively low. However, we want to make a note here that the purpose of autoencoder and latent space in our framework is for better exploitation of the rich and physics-agnostic dataset $D'$ as discussed in the beginning of our response, rather than an architectural improvement.
>
> The reviewer also mentions latent space propagation for reducing inference time for temporal PDE problems. This is indeed an insightful idea. Actually, there are some existing literatures exploring this research directions, such as ***AROMA [Serrano et al., NeurIPS 2024]*** and ***Latent neural operator [Wang et al., NeurIPS 2024]***. However, this is beyond the scope of this work which focuses on developing a training pipeline to better exploit physics-agnostic datasets and enhance operator learning.
>
>
> ***2. Table 6 shows that an AE alone suffices for the AirfRANS dataset, making VAE unnecessary. A similar analysis on the other datasets would help clarify this observation.***
>
> We thank the reviewer for this important suggestion. We agree that including more ablation study on KL weight effect would help clarify the use of VAE instead of a deterministic AE. We have performed an additional ablation experiment for the KL weight effect on the Stress dataset with GNOT being the stage 2 neural operator as shown in the table below. The VAEs are trained on 7600 geometry-only data in stage 1 as shown in Table 3 of the manuscript with GNOT and Transolver trained on 1900 data labeled with stress fields in stage 2.
> ||GNOT|+VAE(0.01)|+VAE(0.001)|+VAE(0.0001)|+AE|
> |:---:|:---:|:---:|:---:|:---:|:---:|
> |Stress, Mesh ($\times 10^{-2}$)|9.8|9.9|**9.0**|9.3|10.3|
> |Stress, Random ($\times 10^{-2}$)|10.3|9.7|8.3|**8.0**|8.9|
>
> ||Transolver|+VAE(0.01)|+VAE(0.001)|+VAE(0.0001)|+AE|
> |:---:|:---:|:---:|:---:|:---:|:---:|
> |Stress, Mesh ($\times 10^{-2}$)|11.5|12.1|**11.2**|11.3|11.7|
> |Stress, Random ($\times 10^{-2}$)|11.5|11.4|9.7|**9.4**|10.4|
>
> It can be observed from the table that using KL weight of 0.001 or 0.0001 for stage 1 pretraining provides better geometry encoding for GNOT on the stress dataset. Although the deterministic autoencoder gives the best geometry representation for Transolver on the AirfRans dataset (Table 6 in the manuscript), training a probabilistic latent space with KL weight of 0.001 or 0.0001 tend to provide more robust performance. Using a purely deterministic autoencoder can sometimes degrade the performance. That said, the optimal choice can depend on the neural operator used in stage 2, other training hyperparameters, and the dataset being learned. Therefore, we pick a consistent KL weight of 0.001 for most experiments in this work to focus on the effectiveness of the two-stage framework rather than detailed parameter tuning.
>
> We have included these results in the updated manuscript.

---

> ### Author Response · Authors · 2025-11-20
>
> ***3. Comparison with other geometry-agnostic methods, such as GINO.***
>
> We thank the reviewer for highlighting this literature. GINO from Li et al. has a very different motivation and research direction from our work. We have briefly introduced GINO's contribution in the Related Work section. In summary, GINO proposed a novel architecture: GNO(graph neural operator)-FNO(fourier neural operator)-GNO to handle operator learning tasks on large scale irregular 3D mesh problems. On the other hand, the focus of our work is to propose the two-stage training framework to better exploit the large physics-agnostic geometry dataset rather than architectural improvement.
>
> Also, we think that GINO has no physics-agnostic geometry encoder (we believe the reviewer meant "physics-agnostic" in the comment, where "geometry-agnostic" is a typo) as the training is fully end-to-end in a supervised manner, where all data are labeled with physical field solutions. That being said, GINO can not learn from physics-agnostic dataset and should be considered orthogonal to our work.
>
> ***4. How to address problems with sharp boundaries and more chaotic dynamics, such as turbulence, with the proposed approach?***
>
> Thank you for these interesting questions.
>
> - Sharp geometric and material boundaries are naturally handled in our framework: in all four datasets, the stage 1 encoder is pretrained with an occupancy proxy task to predict an approximate indicator function of the domain. This proxy explicitly exposes boundary discontinuities such as corners and interfaces to the encoder. If the reviewer is referring to a different notion by "sharp boundaries", we would greatly appreciate clarification.
> - Regarding chaotic dynamics such as turbulence, the AirfRans dataset includes turbulent flows but only through steady RANS solutions (not chaotic in time). A potential future extension would be to evaluate our approach on time dependent chaotic datasets such as those in PDEBench. However, time dependent datasets are considered autoregressive tasks where neural operator learns mapping from $u_{i}$ to $u_{i+1}$ instead of $a$ to $u$. For such tasks, pretraining is physics-aware as the input is readily a physical field, which is beyond the scope of this paper on developing physics-agnostic pretraining strategy.

---

> ### Author Response · Authors · 2025-11-28
>
> As the rebuttal deadline is approaching, we would be very grateful if reviewer En7x could let us know whether our clarifications adequately address your concerns.

---

### Meta-Review · Area_Chair_n2hC · 2026-01-07

**Summary:**

This paper introduces a two-stage operator learning framework that first learns a latent geometric representation using an occupancy prediction model, followed by conditional inference for query locations through a transformer-based neural operator. It has a reasonable motivation by an important industrial scenario that PDE solution data are expensive and thus scarce, but physics-agnostic geometry data are cheap and abundant. It thus provides a physics-agnostic pretraining framework for irregular grids where a VAE is trained on the rich geometry-only data with proxy task to predict occupancy field. Empirical evaluations and ablations on 2D and 3D datasets and three strong neural operators validate the performance gains.

In general, reviewers believe the geometry pre-training framework to be interesting and well-motivated. While the technical roadmap based on VAE is less pronounced due to its expressivity limits on more complex geometries and multiphysics PDEs, also the practicality for sharp boundaries and more chaotic dynamics is not established sufficiently, the potential of the framework has been partially validated. Hence the current work has contributed something desirable by the community. In all, the merits slightly outweigh the flaws. A borderline acceptance.

**Reviewer Concerns:**

Major concerns on the technical part have been addressed well. Some concerns regarding the practicality and deeper understanding of the methods are still outstanding.

**Reviewer Scores:**

Reviewer En7x didn't respond. The listed concerns are valid in the AC's opinion but the authors are only able to partially address these concerns. Therefore, the paper is on verge and rejecting the paper won't be a great loss to the conference.

Reviewer xrAn didn't respond. Reviewer gave a reject recommendation, but the main concerns are less relevant to the neural operator area. So the review and score are down-weighted by the AC.

Reviewer pbnB adjusted the rating to 6 before the major leakage.

---

### Decision · Program_Chairs · 2026-01-26

Accept (Poster)